# Any Large Language Model Can Be a Reliable Judge: Debiasing with a Reasoning-based Bias Detector

**Haoyan Yang[1]***, **Runxue Bao[2]†**, **Cao Xiao[2]**,
**Jun Ma[2]**, **Parminder Bhatia[2]**, **Shangqian Gao[3]†** **Taha Kass-Hout[2]**
[1]New York University    [2]GE Healthcare    [3]Florida State University

## Abstract

LLM-as-a-Judge has emerged as a promising tool for automatically evaluating generated outputs, but its reliability is often undermined by potential biases in judgment. Existing efforts to mitigate these biases face key limitations: in-context learning-based methods fail to address rooted biases due to the evaluator's limited capacity for self-reflection, whereas fine-tuning is not applicable to all evaluator types, especially closed-source models. To address this challenge, we introduce the **R**easoning-based **B**ias **D**etector (RBD), which is a plug-in module that identifies biased evaluations and generates structured reasoning to guide evaluator self-correction. Rather than modifying the evaluator itself, RBD operates externally and engages in an iterative process of bias detection and feedback-driven revision. To support its development, we design a complete pipeline consisting of biased dataset construction, supervision collection, distilled reasoning-based fine-tuning of RBD, and integration with LLM evaluators. We fine-tune four sizes of RBD models, ranging from 1.5B to 14B, and observe consistent performance improvements across all scales. Experimental results on 4 bias types—verbosity, position, bandwagon, and sentiment—evaluated using 8 LLM evaluators demonstrate RBD's strong effectiveness. For example, the RBD-8B model improves evaluation accuracy by an average of 18.5% and consistency by 10.9%, and surpasses prompting-based baselines and fine-tuned judges by 12.8% and 17.2%, respectively. These results highlight RBD's effectiveness and scalability. Additional experiments further demonstrate its strong generalization across biases and domains, as well as its efficiency.[3]

## 1   Introduction

Powered by the strong capabilities of LLMs, LLM-as-a-Judge has emerged as a promising alternative to human evaluation in various NLP tasks [16, 27, 6, 57]. LLM-based evaluation provides a faster and more scalable alternative to human judgment. It is now widely adopted in benchmarking and automated evaluation pipelines. However, despite these advantages, LLM-based evaluation remains imperfect. A key concern is the presence of bias, which can lead to unreliable or unfair assessments [7, 51, 46, 26]. Our findings further confirm that even state-of-the-art models GPT-4o [35] and Claude-3.5-sonnet [3] consistently exhibit detectable biases. Therefore, addressing these biases is essential to improving the transparency and credibility of LLM-based evaluation.

To mitigate these biases, recent work has proposed two main strategies. Some approaches use in-context learning (ICL), prompting the LLM evaluator with carefully crafted instructions and

---

*Work was done during the internship at GE Healthcare.

†Correspondence to: Runxue Bao, Shangqian Gao (`baorunxue@gmail.com`, `sgao@cs.fsu.edu`)

[3]All data and code are available at `https://github.com/Joyyang158/Reasoning-Bias-Detector`.

39th Conference on Neural Information Processing Systems (NeurIPS 2025).

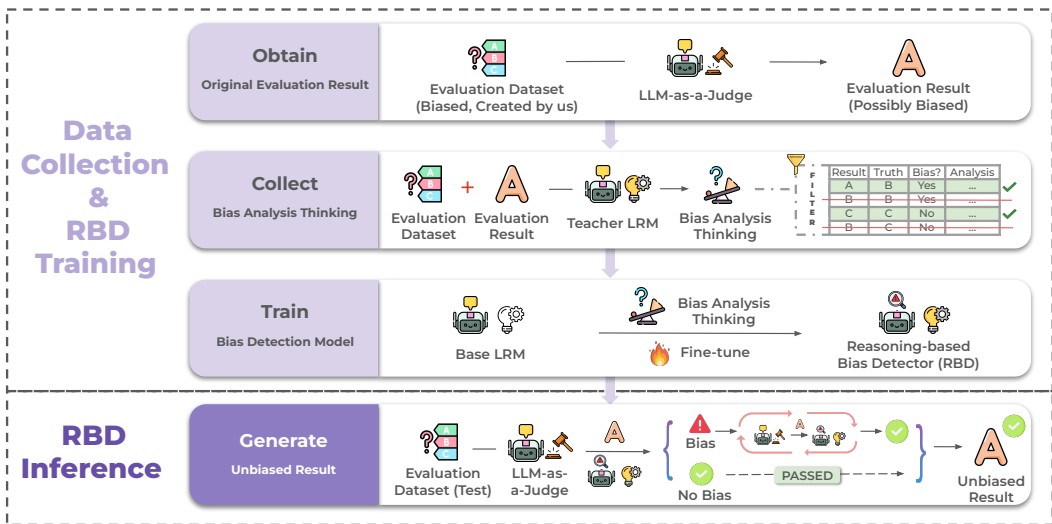

Figure 1: Overview of the Reasoning-based Bias Detector (RBD) framework. During **RBD inference**, it examines biased evaluation results produced by an LLM-as-a-Judge. If bias is identified, RBD generates a reasoning-based bias analysis to guide the LLM in reflecting on and potentially revising its evaluation; otherwise, the original judgment remains unchanged. To train RBD, we design a **data collection and distilled reasoning-based training pipeline**. We first construct a biased dataset containing specific types of bias and collect possibly biased evaluation results from the LLM evaluator. Then, a teacher Language Reasoning Model (LRM) produces bias analysis thinking based on the evaluation context. These analyses are filtered and used to fine-tune a base LRM into the final RBD model capable of identifying and correcting evaluation bias.

illustrative examples to promote more deliberate, reflective judgments and reduce potential biases [34, 49, 9, 11, 42]. Others fine-tune language models using evaluation-style corpora—consisting of prompts, model outputs, and human or LLM-generated preferences—to improve their capabilities in ranking or scoring tasks, effectively training an LLM evaluator [59, 28, 21, 48, 22, 41]. However, both strategies exhibit significant limitations. Prompting-based methods are easy to implement and broadly applicable, but some deep-rooted biases remain difficult to mitigate [56, 33]—especially in weaker models [50]—as surface-level instructions are insufficient to alter the model's underlying behavior. Fine-tuning approaches cannot be applied to closed-source models, which are widely used in LLM-as-a-Judge applications. In addition, they require large-scale, high-quality preference data, and the models may overfit to evaluation-specific patterns, thereby reducing their generality across tasks.

To address these gaps, we propose a new approach that introduces a Reasoning-based Bias Detector (RBD) to identify potential biases in LLM evaluation and generate reasoning-based analyses to assist the evaluator in self-reflection. Instead of directly modifying or fine-tuning the evaluator, RBD serves as a companion module that inspects the evaluation output, determines whether it is biased, and provides structured reasoning as a reference to encourage more accurate and fair reassessments. This design is applicable to both open- and closed-source LLM evaluators, enhancing evaluation reliability through targeted and interpretable reasoning feedback. To train and evaluate this RBD module, we construct a comprehensive framework as shown in Figure 1 illustrates the complete bias mitigation pipeline, including four key stages: constructing targeted biased evaluation datasets, collecting high-quality reasoning-based annotations, performing distilled reasoning-based fine-tuning of the RBD, and integrating it with the evaluator for iterative evaluation. As demonstrated in Appendix A, we focus on four representative structural biases in LLM evaluators: verbosity bias [46, 57, 37, 55], position bias [40, 52, 30], bandwagon bias [23, 39], and sentiment bias [25, 51, 15], which are commonly observed across tasks. These types of bias reflect systematic evaluation flaws not tied to specific topics or domains, making them especially important to detect and mitigate. In summary, our contributions are:

**Reasoning-based Bias Detector (RBD) for LLM Evaluator**   We introduce a modular RBD that interacts with LLM evaluators to detect potential biases in their judgments and generate structured

reasoning to support self-reflection and correction of LLM evaluators. RBD integrates seamlessly with diverse LLM evaluators, consistently enhancing evaluation accuracy and reliability.

**Scalable Training and Evaluation Pipeline**   We develop an end-to-end pipeline comprising bias dataset construction, reasoning corpus generation, and model training. This includes four curated bias-specific datasets (0.5k instances each) and 1.67k reasoning traces from a teacher Language Reasoning Model (LRM), which are used to fine-tune RBD models spanning 1.5B to 14B parameters.

**Empirical Validation of Effectiveness, Generality, and Efficiency**   Extensive experiments across four bias types and eight evaluator configurations demonstrate robust performance gains. RBD-8B, for instance, achieves average improvements of 18.5% in accuracy and 10.9% in consistency. Moreover, RBD generalizes well across domains with low latency and inference cost.

## 2   Related Work

Bias in LLM-based evaluation has been studied via prompt-based and fine-tuned methods. Prompt-based approaches mitigate bias by carefully designing prompts to guide more reliable evaluations, including techniques like instruction reformulation [58, 18, 17, 49, 44, 38, 11]. Further improvements have been made through multi-turn interaction and multi-agent collaboration, which encourage deliberation and reduce individual judgment bias [4, 5, 53]. Moreover, fine-tuned methods directly train evaluator models on curated preference data to learn de-biased decision patterns [31, 45, 20, 29, 59, 41, 21, 22, 48]. Details of these works are provided in Appendix B.

Beyond these efforts, LRMs have recently been explored as tools for improving LLM-based evaluation [19, 47]. However, instead of directly fine-tuning LRMs to act as evaluators, we proposes a novel framework that fine-tunes LRMs as bias detectors, which collaborate with LLM evaluators to reflect on and revise potentially biased decisions. This setup enhances evaluation performance and applicability without requiring access to or modification of the LLM evaluator.

## 3   Biased Dataset Construction and LLM-as-a-Judge Evaluation

Table 1: Base datasets used to construct the original and biased datasets. GSM8K is a math QA dataset with reasoning and final answers; Arena contains AI-generated chat instruction pairs; and ScienceQA includes multimodal multiple-choice science questions.

| Bias Type | Base Dataset |
|---|---|
| Verbosity Bias | GSM8K [10] |
| Position Bias | Arena [8] |
| Bandwagon Bias | Arena [8] |
| Sentiment Bias | ScienceQA [32] |

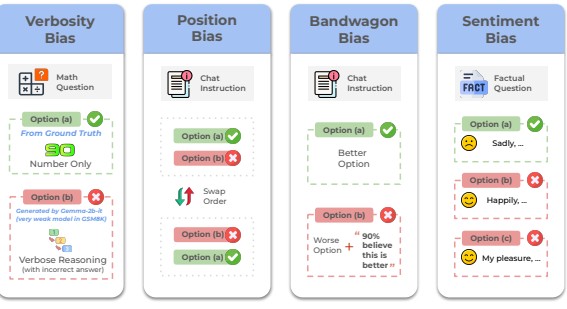

Figure 2: Overview of the bias dataset construction, illustrating how we create the specific biased dataset for each bias (Verbosity, Position, Bandwagon, Sentiment).

For each type of bias, we construct an unbiased dataset $\mathcal{D}$ and its biased counterpart $\mathcal{D}_{\text{bias}}$ based on the corresponding base dataset listed in Table 1. Both $\mathcal{D}$ and $\mathcal{D}_{\text{bias}}$ are choice-based in format. Each example in $\mathcal{D}_{\text{bias}}$ shares the same question or prompt as in $\mathcal{D}$, but the answer options are modified to introduce the specific bias. As shown in Figure 2, we specifically introduce construction methods for each type of bias. We use the term *positive option* to denote the preferred choice, and refer to the others as *negative option(s)*. Examples of $\mathcal{D}$ and $\mathcal{D}_{\text{bias}}$ are provided in Appendix D.1.

**Verbosity Bias (Pairwise)**   In $\mathcal{D}$, we use the ground truth from the base dataset as the positive option, formatted as ''`<a piece of math reasoning> #### <final answer>`''. We use responses with the same format generated by **Gemma-2B-it** [14] as negative options, which is based on our observation that this model consistently performs poorly on the base dataset and produces incorrect outputs. The prompting setup is provided in Appendix E.1. In $\mathcal{D}_{\text{bias}}$, we use only the ''`<final answer>`'' from the ground truth as the positive option, while keeping the negative option unchanged.

Compared to $\mathcal{D}$ where the option lengths are relatively close (average token lengths: 103 vs. 139), the positive option in $\mathcal{D}_{\text{bias}}$ is significantly shorter (2 vs. 139). This setup introduces a clear verbosity bias in $\mathcal{D}_{\text{bias}}$, as the correct answer is noticeably shorter.

**Position Bias (Pairwise)**    We retain the original option pairs from the base dataset in $\mathcal{D}$, as the base dataset itself is pairwise in nature. In $\mathcal{D}_{\text{bias}}$, we introduce position bias by simply swapping the order of the two options.

**Bandwagon Bias (Pairwise)**    The options in $\mathcal{D}$ remain the same as those in the base dataset. In $\mathcal{D}_{\text{bias}}$, We introduce bandwagon bias by inserting a fabricated majority opinion for the LLM evaluator's reference—*90% of people believe that Option x is better*—where *Option x* is the negative option.

**Sentiment Bias (Multiple-choice, # options = 3 or 4)**    We construct $\mathcal{D}$ by selecting QA pairs from the base dataset that are text-only and contain more than two options. To create $\mathcal{D}_{\text{bias}}$, we use GPT-4o [35] (prompt provided in Appendix E.2) to rewrite the tone of each option without altering its semantic meaning. Specifically, we assign a negative tone (e.g., sad, frustrated) to the positive option and positive tones (e.g., happy, enthusiastic) to the negative options, thereby introducing sentiment bias.

In summary, we construct 4 pair ($\mathcal{D}$ and $\mathcal{D}_{\text{bias}}$) of datasets targeting 4 types of bias across multiple domains, covering both pairwise and multiple-choice formats. Bias is defined to exist when a correct evaluation result in $\mathcal{D}$ becomes incorrect in $\mathcal{D}_{\text{bias}}$. We define a binary variable $b_i$ that indicates whether the evaluation result for the $i$-th example is biased, which can be represented as:

$$b_i = \begin{cases} \texttt{Yes}, & \text{if } \hat{y}_i = y_i \text{ and } \hat{y}_i^{\text{bias}} \neq y_i \\ \texttt{No}, & \text{otherwise} \end{cases} \tag{1}$$

where $\hat{y}_i$ is the prediction in $\mathcal{D}$, $\hat{y}_i^{\text{bias}}$ is the prediction in $\mathcal{D}_{\text{bias}}$, and $y_i$ is the ground truth label. In short, if $b_i = \texttt{Yes}$, it indicates that a sample originally evaluated correctly in $\mathcal{D}$ becomes wrong when evaluated in the biased dataset $\mathcal{D}_{\text{bias}}$.

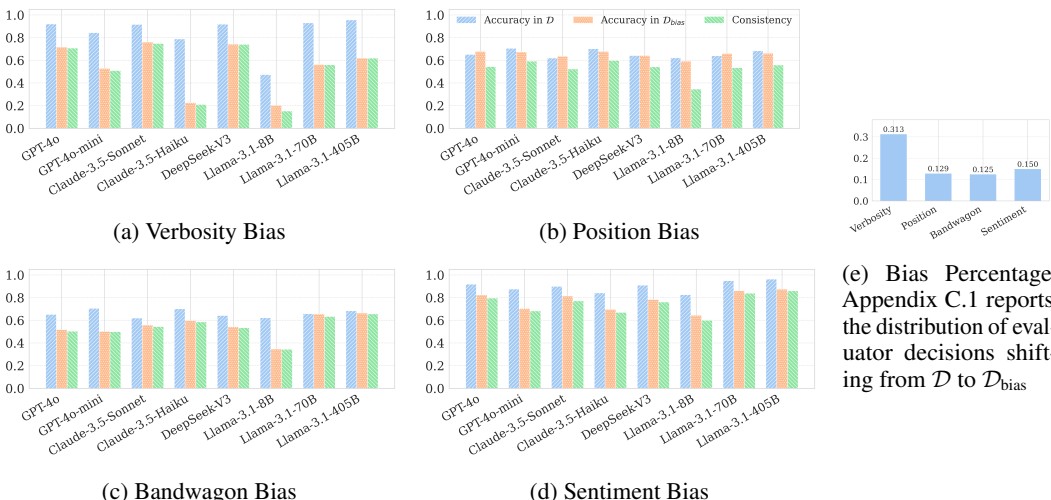

(a) Verbosity Bias        (b) Position Bias

(c) Bandwagon Bias        (d) Sentiment Bias

(e) Bias Percentage. Appendix C.1 reports the distribution of evaluator decisions shifting from $\mathcal{D}$ to $\mathcal{D}_{\text{bias}}$

Figure 3: Performance comparison across four types of bias in $\mathcal{D}$ and $\mathcal{D}_{\text{bias}}$. (a)–(d) show accuracy and consistency drops for each bias type. (e) summarizes the percentage of biased examples.

To assess whether LLM evaluators exhibit these biases, we compare their performance on $\mathcal{D}$ and $\mathcal{D}_{\text{bias}}$. Significant performance drop from $\mathcal{D}$ to $\mathcal{D}_{\text{bias}}$ indicates the presence of bias. As shown in Figure 3a - 3d, we evaluate 8 widely used LLM evaluators to assess whether they exhibit bias on our constructed datasets. These include: (1) OpenAI's `gpt-4o` [35] and `gpt-4o-mini` [36], (2) Anthropic's `claude-3-5-sonnet` [3] and `claude-3-5-haiku` [2], (3) DeepSeek's `DeepSeek-V3` [13], and (4) Meta's LLaMA 3.1 series [1] (8B, 70B, 405B). This selection ensures broad coverage across model sizes, service providers, and both open-source and closed-source evaluators. To evaluate the performance, we report two metrics: (1) **Accuracy**: the percentage of examples where the evaluator selects the correct (preferred) option. (2) **Consistency**: the percentage of examples where the evaluator gives the same correct prediction before and after bias is introduced. Formally, we

define: Accuracy $= \frac{1}{N} \sum_{i=1}^{N} \mathbb{1}\left[\tilde{y}_i = y_i\right]$, Consistency $= \frac{1}{N} \sum_{i=1}^{N} \mathbb{1}\left[\hat{y}_i = y_i \wedge \hat{y}_i^{\text{bias}} = y_i\right]$ where $\tilde{y}_i \in \{\hat{y}_i, \hat{y}_i^{\text{bias}}\}$.

All eight LLM evaluators show a clear drop in both accuracy and consistency when moving from $\mathcal{D}$ to $\mathcal{D}_{\text{bias}}$ across all four bias types, except for accuracy under position bias, which remains stable due to uncertain output position preference, while the consistency drop still indicates bias. As shown in Figure 3e, we computed the distribution of biased behaviors across all four bias types. Verbosity bias causes the most degradation, with 31.3% of examples exhibiting biased behavior, followed by sentiment (15.0%), position (12.9%), and bandwagon bias (12.5%).

## 4  RBD Inference and Training Pipeline

In this section, we first describe the collaborative de-biased evaluation procedure using the LLM evaluator and trained RBD (denoted as $M_J$ and $M_R$), and then provide the training details of RBD.

**Collaborative De-biased Evaluation with RBD**  As shown in Algorithm 1, we iteratively refine the evaluator's judgment with the assistance of RBD to reduce potential bias. The process begins by obtaining an initial evaluation result from an LLM evaluator on an input sample that may exhibit bias. RBD then produces a structured reasoning trace along with a bias prediction to guide the revision process. The output follows the format "`<think> {reasoning trace} </think>` `{bias label}`", where the label indicates whether the initial judgment is considered biased. If no bias is detected, the judgment is accepted as final. Otherwise, the evaluator is prompted again with the generated reasoning trace and label by RBD as additional reference to reflect its decision. This process continues iteratively until RBD confirms there is no bias or the maximum number of iterations is reached.

---

**Algorithm 1:** RBD Inference with LLM Evaluators

---

**Input:** Input $x^{\text{bias}}$, Information of evaluator $I_{\mathcal{M}_J}$, Information of biases $I_{\text{bias}}$, Max iteration $T$
**Output:** Final judgment result $\hat{y}^{\text{final}}$
**Models:** Evaluator $\mathcal{M}_J$, RBD $\mathcal{M}_R$;
$\hat{y}^{\text{bias}} \leftarrow \mathcal{M}_J(x^{\text{bias}})$;
**for** $t \leftarrow 1$ **to** $T$ **do**
    $\hat{y}^r \leftarrow \mathcal{M}_R(x^{\text{bias}}, \hat{y}^{\text{bias}}, I_{\mathcal{M}_J}, I_{bias})$;
    $\hat{b} \leftarrow$ Split $(\hat{y}^r)$ with </think> token;
    **if** $\hat{b} == No$ **then**
        $\hat{y}^{\text{final}} \leftarrow \hat{y}^{\text{bias}}$;
        **break**;
    $\hat{y}^{\text{bias}} \leftarrow \mathcal{M}_J(x^{\text{bias}}, \hat{y}^r)$;
**return** $\hat{y}^{\text{final}}$

---

**Reasoning Data Collection and RBD Training** For one example $x_i^{\text{bias}}$ in $\mathcal{D}$, we begin by applying the LLM evaluator $\mathcal{M}_J$ to obtain its evaluation result: $\hat{y}_i^{\text{bias}} = \mathcal{M}_J(x_i^{\text{bias}})$. By comparing $\hat{y}_i^{\text{bias}}$ with $\hat{y}_i$ and $y_i$, we derive a bias label $b_i$ indicating whether the evaluator's output is biased as shown in Eq. 1. Each data point is associated with a specific bias type $t_i \in \{\text{verbosity}, \text{position}, \text{bandwagon}, \text{sentiment}\}$, resulting in a data tuple of the form $(x_i^{\text{bias}}, \hat{y}_i^{\text{bias}}, b_i, t_i)$.

To construct high-quality reasoning data for RBD training, we prompt a teacher model $\mathcal{M}_T$ to analyze each data tuple. The prompt includes the biased input example, the bias type to consider, the name of the evaluator model, which implicitly informs the teacher model of the evaluator's capability, and its corresponding evaluation result. The teacher model then generates a structured reasoning output: $y_i^r = \mathcal{M}_T(x_i^{\text{bias}}, t_i, I_{\mathcal{M}_J}, \hat{y}_i^{\text{bias}})$ where $y_i^r$ follows the

format `<think>`$\hat{r}_i$`</think>`$\hat{b}_i$, with $\hat{r}_i$ denoting the reasoning trace and $\hat{b}_i \in \{\texttt{Yes}, \texttt{No}\}$ indicating the predicted bias label, and $I_{\mathcal{M}_J}$ is defined in Algorithm 1 (prompt template of $M_T$ used to generate $y_i^r$ can be found in Appendix E.3). The reasoning trace typically includes three parts: (1) identification of the potential bias type; (2) a paragraph of comparative analysis that explicitly compares the given options and supports the predicted bias label based on the bias definition (3) an assessment of how the evaluator's capabilities may influence its susceptibility to the identified bias. To ensure reliability, we retain only those instances where the teacher-predicted bias label matches the ground-truth bias label obtained before: $\mathcal{D}_{\text{train}} = \left\{ (x_i^{\text{bias}}, \hat{y}_i^{\text{bias}}, y_i^r, I_{\mathcal{M}_J}, t_i) \,\middle|\, \hat{b}_i = b_i \right\}$.

We fine-tune a base language reasoning model $\mathcal{M}_L$ into a RBD $\mathcal{M}_R$ using $\mathcal{D}_{\text{train}}$. Notably, to enhance the robustness and generalization ability of RBD, we train it jointly on examples from all four types of bias, rather than training separate models for each bias type. The training objective is to maximize the likelihood of generating $y_i^r$ given $x_i^{\text{bias}}, \hat{y}_i^{\text{bias}}, I_{\mathcal{M}_J}$ and $I_{\text{bias}}$, as defined in Algorithm 1 to describe definitions for all biases (prompt template for RDB training can be found in Appendix E.4):

$$\min_{\theta_L} \mathcal{L}(\theta_L) := -\log P(y_i^r \mid x_i^{\text{bias}}, \hat{y}_i^{\text{bias}}, I_{\mathcal{M}_J}, I_{\text{bias}})$$

where $\theta_L$ is the parameters of $\mathcal{M}_L$. After fine-tuning, the generated output $\hat{y}_i^r$ can be applied to guide the LLM evaluator for the reference to detect the potential bias during evaluation.

# 5 Experiments and Analysis

## 5.1 Experimental Setup

**RBD Model Size Series** We develop 4 RBD variants with different model sizes: 1.5B, 7B, 8B, and 14B. The corresponding base LRMs are `DeepSeek-R1-Distill-Qwen-1.5B`, `Qwen-7B`, `LLaMA-8B`, `Qwen-14B` [12]. Details of the RBD training setup and loss curves can be found in Appendix C.2.

**Dataset and Metric** (1) *RBD Fine-tuning and Evaluation:* $|\mathcal{D}_{\text{train}}|$ is 1.67k. Detailed statistics are provided in Appendix C.3. We additionally construct a test set of 0.5k examples to evaluate the RBD fine-tuning performance, with an approximately equal distribution across four bias types. For each bias type, the bias labels are balanced with a 50:50 ratio of `Yes` and `No`. For evaluation, we report Accuracy, Precision (for the `Yes` class), Recall (for the `Yes` class), and F1 Score based on the prediction of bias labels. (2) *LLM Evaluator with RBD:* For each bias type, $|\mathcal{D}|$ and $|\mathcal{D}_{\text{bias}}|$ are both 0.5k. We evaluate the performance gain of LLM-based evaluators with and without RBD to validate its effectiveness. The LLM evaluators and performance metrics used are the same as those in the bias existence analysis described in Section 3.

## 5.2 Performance of RBD Fine-tuning

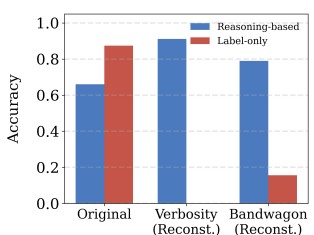

Figure 4: Comparison of reasoning-based and label-only fine-tuning on the original test set and two diagnostic sets.

Firstly, to verify that the superior performance of RBD does not stem from merely hacking synthetic dataset artifacts, we compare it with an alternative approach, bias-only fine-tuning. In that setting, the model is trained solely with bias labels using a binary classification head, without any reasoning supervision. As shown in Figure 4, we find that while bias-only fine-tuning can achieve high accuracy on the original test set, it overfits to superficial patterns and lacks true bias understanding. Specifically, it fails catastrophically when input formats change. On two diagnostic sets (examples can be found in Appendix D.2): (1) a reconstructed verbosity set from GSM8K, where longer answers are always correct, and (2) a reconstructed bandwagon set from Arena, where the majority opinion is always correct. The bias-only model misclassifies most examples, even dropping to zero accuracy on verbosity, revealing its reliance on shallow format cues (e.g., preferring shorter or minority responses). In contrast, the reasoning-based fine-tuning remains robust and consistent across these shifts, confirming that RBD enables genuine bias reasoning rather than exploiting dataset-specific patterns.

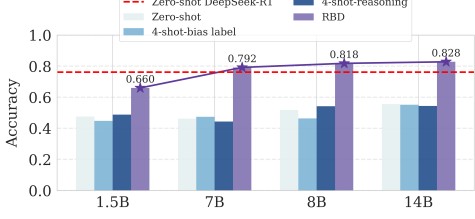
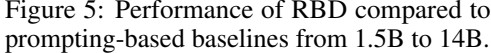

Figure 5: Performance of RBD compared to prompting-based baselines from 1.5B to 14B.

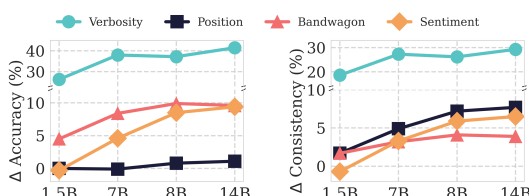

Figure 6: Avg. performance gains of 8 LLM evaluators with RBDs (1.5B–14B) across 4 bias types.

In addition, we compare each RBD model with its corresponding base LRM of the same size (e.g., RBD-1.5B vs. DeepSeek-R1-Distill-Qwen-1.5B) under three prompting settings—zero-shot, 4-shot with bias labels, and 4-shot with reasoning, and additionally include the zero-shot `DeepSeek-R1` as a reference baseline. As shown in Figure 5, RBD consistently outperforms all baselines across

Table 2: Bias-specific evaluation of LLM evaluators with and without **RBD-8B**. The table reports results on 4 bias types (verbosity, position, bandwagon, and sentiment) across 8 LLM evaluators. For each evaluator, the *First Row* reports its *Accuracy* performance, while the *Second Row* shows its *Consistency* performance.

| | Verbosity | Position[4] | Bandwagon | Sentiment |
|---|---|---|---|---|
| **GPT-4o** | $0.716 \rightarrow 0.912^{(+19.6\%)}$ | $0.665 \rightarrow 0.653$ | $0.518 \rightarrow 0.588^{(+7.0\%)}$ | $0.824 \rightarrow 0.852^{(+2.8\%)}$ |
| | $0.708 \rightarrow 0.866^{(+15.8\%)}$ | $0.544 \rightarrow 0.610^{(+6.6\%)}$ | $0.504 \rightarrow 0.536^{(+3.2\%)}$ | $0.796 \rightarrow 0.816^{(+2.0\%)}$ |
| **GPT-4o-mini** | $0.528 \rightarrow 0.854^{(+32.6\%)}$ | $0.690 \rightarrow 0.692$ | $0.502 \rightarrow 0.634^{(+13.2\%)}$ | $0.705 \rightarrow 0.833^{(+12.8\%)}$ |
| | $0.510 \rightarrow 0.756^{(+24.6\%)}$ | $0.592 \rightarrow 0.636^{(+4.4\%)}$ | $0.500 \rightarrow 0.588^{(+8.8\%)}$ | $0.683 \rightarrow 0.771^{(+8.8\%)}$ |
| **Claude-3.5-sonnet** | $0.760 \rightarrow 0.964^{(+20.4\%)}$ | $0.628 \rightarrow 0.632$ | $0.558 \rightarrow 0.670^{(+11.2\%)}$ | $0.814 \rightarrow 0.872^{(+5.8\%)}$ |
| | $0.748 \rightarrow 0.896^{(+14.8\%)}$ | $0.524 \rightarrow 0.594^{(+7.0\%)}$ | $0.544 \rightarrow 0.572^{(+2.8\%)}$ | $0.772 \rightarrow 0.814^{(+4.2\%)}$ |
| **Claude-3.5-haiku** | $0.226 \rightarrow 0.870^{(+64.4\%)}$ | $0.690 \rightarrow 0.685$ | $0.596 \rightarrow 0.688^{(+9.2\%)}$ | $0.695 \rightarrow 0.830^{(+13.5\%)}$ |
| | $0.210 \rightarrow 0.710^{(+50.0\%)}$ | $0.598 \rightarrow 0.639^{(+4.1\%)}$ | $0.586 \rightarrow 0.622^{(+3.6\%)}$ | $0.669 \rightarrow 0.760^{(+9.1\%)}$ |
| **Deepseek-V3** | $0.742 \rightarrow 0.934^{(+19.2\%)}$ | $0.642 \rightarrow 0.662$ | $0.542 \rightarrow 0.622^{(+8.0\%)}$ | $0.784 \rightarrow 0.826^{(+4.2\%)}$ |
| | $0.740 \rightarrow 0.876^{(+13.6\%)}$ | $0.542 \rightarrow 0.626^{(+8.4\%)}$ | $0.534 \rightarrow 0.562^{(+2.8\%)}$ | $0.762 \rightarrow 0.794^{(+3.2\%)}$ |
| **LLaMA-3.1-8B** | $0.202 \rightarrow 0.920^{(+71.8\%)}$ | $0.607 \rightarrow 0.624$ | $0.346 \rightarrow 0.540^{(+19.4\%)}$ | $0.644 \rightarrow 0.866^{(+22.2\%)}$ |
| | $0.152 \rightarrow 0.440^{(+28.8\%)}$ | $0.346 \rightarrow 0.434^{(+8.8\%)}$ | $0.344 \rightarrow 0.442^{(+9.8\%)}$ | $0.600 \rightarrow 0.752^{(+15.2\%)}$ |
| **LLaMA-3.1-70B** | $0.562 \rightarrow \mathbf{0.940}^{(+37.8\%)}$ | $0.650 \rightarrow 0.668$ | $0.656 \rightarrow 0.710^{(+5.4\%)}$ | $0.861 \rightarrow \mathbf{0.900}^{(+3.9\%)}$ |
| | $0.560 \rightarrow 0.894^{(+33.4\%)}$ | $0.534 \rightarrow 0.614^{(+8.0\%)}$ | $0.634 \rightarrow 0.644^{(+1.0\%)}$ | $0.838 \rightarrow 0.869^{(+3.1\%)}$ |
| **LLaMA-3.1-405B** | $0.619 \rightarrow 0.937^{(+31.8\%)}$ | $0.673 \rightarrow 0.690$ | $0.663 \rightarrow \mathbf{0.719}^{(+5.6\%)}$ | $0.874 \rightarrow 0.899^{(+2.5\%)}$ |
| | $0.619 \rightarrow \mathbf{0.902}^{(+28.3\%)}$ | $0.558 \rightarrow \mathbf{0.664}^{(+10.6\%)}$ | $0.657 \rightarrow \mathbf{0.663}^{(+0.6\%)}$ | $0.860 \rightarrow \mathbf{0.879}^{(+1.9\%)}$ |
| **AVERAGE** | +37.2% | +7.2% | +9.9% | +8.5% |
| | +26.2% | | +4.1% | +5.9% |

model scales. Notably, it surpasses zero-shot DeepSeek-R1 starting from RBD-7B and achieves the highest accuracy of 0.828 on RBD-14B. These results demonstrate the effectiveness and scalability of reasoning-based fine-tuning. Full metrics are provided in Appendix C.4.

## 5.3 Performance of LLM Evaluators with RBD

In this part, we evaluate the benefits of integrating RBD into LLM evaluators. As shown in Table 2, adding the RBD-8B module consistently improves performance across all four bias types and all LLM evaluators. The most significant accuracy gain is observed for verbosity bias. The performance boost is especially notable for smaller models—for instance, LLaMA-3.1-8B sees an improvement of up to 71.8% on verbosity bias. These results broadly demonstrate that RBD provides an external reasoning mechanism that helps LLM evaluators enhance their judgment and reliability. The prompt for invoking LLM evaluators and an end-to-end example showing how RBD assists the evaluator are provided in Appendices D.3 and E.5. We further investigate how RBD performance scales with model size. As shown in Figure 6, larger RBD leads to greater improvements in LLM evaluation. Specifically, as RBD size increases from 1.5B to 14B, the average accuracy gain rises from +6.5% to +15.4%, while consistency improves from +5.3% to +11.9%. These findings demonstrate the scalability of RBD. Detailed performance results for each LLM evaluator with RBD other than 8B are provided in Appendix C.5.

## 5.4 Generalization Ability of RBD

**Prompt robustness** We evaluate RBD's robustness to prompt variations by removing the evaluator model name and the specific bias type. As detailed in Appendix C.6, RBD achieves comparable performance under these ablated settings, demonstrating that it can generalize well without relying on prompt-specific cues.

---

[4]For Position Bias, we focus primarily on consistency, as RBD is designed to identify inconsistencies when the order of options is swapped, rather than altering the evaluator's inherent ability to judge correctness.

**Multi-bias robustness**   We test whether RBD can handle multiple biases simultaneously by augmenting the verbosity dataset with a bandwagon bias, where 90% of samples prefer longer answers. As shown in Figure 7, smaller models like `GPT-4o-mini` and `Claude-3.5-haiku` degrade sharply under this multi-bias setup, while larger models remain stable. Applying RBD yields substantial recovery—Figure 8 shows +45% and +63.8% gains for the two models, demonstrating RBD's robustness even when multiple bias types interact (an illustrative example is provided in Appendix D.4).

**Cross-domain generalization**   To verify that RBD generalizes beyond its training domain, we apply it to a new factual QA dataset (FactQA) under verbosity bias. Although RBD was trained exclusively on math problems, it remains effective in this factual setting. As shown in Figure 9, using `Claude-3.5-haiku` as the evaluator, RBD-8B and RBD-14B improve accuracy from 0.712 to 0.796 and 0.882, and consistency from 0.654 to 0.694 and 0.740, respectively (the construction setting and an example of FactQA are provided in Appendix D.5).

**External benchmark evaluation**   Finally, we evaluate RBD on two external benchmarks, LLMBar [54] and JudgeBench [43], covering two bias types (verbosity and bandwagon). The experimental settings are described in Appendix C.7 in detail. As shown in Table 3, RBD consistently improves performance, raising averages from 0.681→0.792 and 0.705→0.773 for verbosity bias, and from 0.694→0.765 and 0.578→0.624 for bandwagon bias across JudgeBench and LLMBar. These results confirm that RBD not only performs well within synthetic datasets but also generalizes robustly to external, unseen benchmarks.

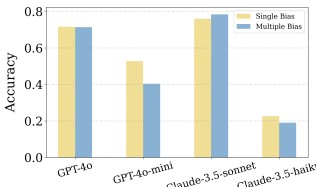

Figure 7: Comparison of accuracy for GPT and Claude model families under single bias and multiple bias (Verbosity + Bandwagon).

Figure 8: Accuracy improvements for GPT-4o-mini and Claude-3.5-haiku with RBD-8B under single and multiple bias.

Figure 9: Evaluation on the FactQA dataset using Claude-3.5-Haiku with RBD-8B and RBD-14B to test RBD's cross-domain generalization in mitigating verbosity bias.

## 5.5   Comparison to Existing Approaches

We further compare our method against existing prompting-based and fine-tuned judge approaches by using LLaMA-3.1-70B as the LLM evaluator with RBD-8B. For prompting-based baselines, we use an in-context learning setup (prompt is in Appendix E.7) that explicitly instructs the model to avoid specific biases. For fine-tuned judges, we compare against two strong models—Prometheus2-8×7B [22] and Skywork-LLaMA-3.1-70B [41]—both of which perform well on the JUDGEBENCH benchmark [43]. As shown in Table 4, our method outperforms both prompting-based and fine-tuned judge baselines. Prometheus performs poorly especially on bandwagon bias (0.014), likely due to poor generalization and sensitivity to majority-preference patterns. These results show that RBD not only achieves superior performance but also offers strong generalization and flexibility, enabling easy integration into existing LLM evaluators without the need for end-to-end fine-tuning.

## 6   Analysis and Discussion

**Importance of Reasoning in De-biased Inference**   In Section 5.2, we showed that reasoning is essential for RBD training. We now further evaluate its role during inference. As shown in Table 5, using full reasoning instead of a binary label consistently improves GPT-4o's performance across all bias types. These results highlight the importance of reasoning in guiding evaluators toward more robust and bias-aware decisions during inference.

Table 3: Performance of LLM evaluators with and without RBD on LLMBar [54] and JudgeBench [43] benchmarks across two bias types.

| Bias Type | LLM Evaluator | LLMBar | | | JudgeBench | | |
|---|---|---|---|---|---|---|---|
| | | w/o RBD | w/ RBD-8B | w/ RBD-14B | w/o RBD | w RBD-8B | w/RBD-14B |
| Verbosity | GPT-4o | 0.798 | 0.798 | 0.820 | 0.723 | 0.734 | 0.747 |
| | GPT-4o-mini | 0.685 | 0.663 | 0.775 | 0.687 | 0.740 | 0.805 |
| | Claude-3.5-sonnet | 0.730 | 0.753 | 0.787 | 0.711 | 0.747 | 0.773 |
| | Claude-3.5-haiku | 0.427 | 0.472 | 0.708 | 0.651 | 0.753 | 0.740 |
| | DeepSeek-V3 | 0.798 | 0.798 | 0.854 | 0.699 | 0.684 | 0.753 |
| | LLaMA-3.1-8B | 0.539 | 0.596 | 0.798 | 0.639 | 0.671 | 0.759 |
| | LLaMA-3.1-70B | 0.719 | 0.685 | 0.795 | 0.723 | 0.740 | 0.785 |
| | LLaMA-3.1-405B | 0.753 | 0.764 | 0.798 | 0.807 | 0.741 | 0.821 |
| | AVERAGE | 0.681 | 0.691 | **0.792** | 0.705 | 0.726 | **0.773** |
| Bandwagon | GPT-4o | 0.780 | 0.790 | 0.830 | 0.580 | 0.620 | 0.610 |
| | GPT-4o-mini | 0.650 | 0.670 | 0.770 | 0.530 | 0.530 | 0.640 |
| | Claude-3.5-sonnet | 0.770 | 0.760 | 0.800 | 0.640 | 0.640 | 0.610 |
| | Claude-3.5-haiku | 0.530 | 0.580 | 0.670 | 0.540 | 0.560 | 0.580 |
| | DeepSeek-V3 | 0.740 | 0.760 | 0.780 | 0.690 | 0.670 | 0.700 |
| | LLaMA-3.1-8B | 0.540 | 0.590 | 0.670 | 0.530 | 0.580 | 0.570 |
| | LLaMA-3.1-70B | 0.730 | 0.770 | 0.770 | 0.570 | 0.600 | 0.690 |
| | LLaMA-3.1-405B | 0.810 | 0.790 | 0.830 | 0.540 | 0.560 | 0.590 |
| | AVERAGE | 0.694 | 0.714 | **0.765** | 0.578 | 0.595 | **0.624** |

Table 4: Bias evaluation results comparing our method (RBD-8B + LLaMA-3.1-70B) with Prompt-based and fine-tuned judge—Prometheus2-8×7B (Prometheus) [22] and Skywork-LLaMA-3.1-70B (Skywork) [41]. Accuracy is used, except consistency for position.

| Bias Type | Vanilla | Prompt | Prometheus | Skywork | Ours |
|---|---|---|---|---|---|
| Verbosity | 0.562 | 0.576 | 0.226 | 0.592 | **0.940** |
| Position | 0.534 | 0.558 | 0.394 | 0.606 | **0.614** |
| Bandwagon | 0.656 | 0.664 | 0.014 | 0.664 | **0.710** |
| Sentiment | 0.861 | 0.856 | 0.362 | 0.614 | **0.900** |

Table 5: Comparison of providing only the **bias label** v.s. the full **reasoning** trace from RBD-8B during inference. GPT-4o is the evaluator. Accuracy is used, except consistency for position.

| Bias Type | Original | Bias Label | Reasoning |
|---|---|---|---|
| Verbosity | 0.716 | 0.764 | **0.912** |
| Position | 0.544 | 0.566 | **0.610** |
| Bandwagon | 0.518 | 0.536 | **0.590** |
| Sentiment | 0.824 | 0.832 | **0.852** |

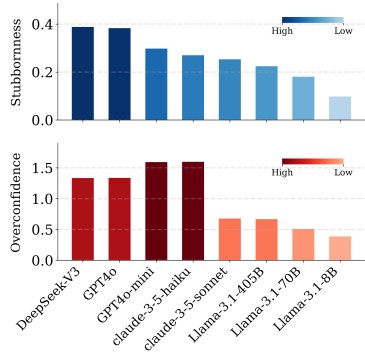

Figure 10: Comparison of LLM evaluators in terms of stubbornness (top) and overconfidence (bottom). Stubbornness reflects how often a model sticks to its answer despite bias warnings; overconfidence measures how often such answers are wrong.

**Stubbornness and Overconfidence Analysis of LLM Evaluators with RBD** We introduce two behavioral metrics to analyze LLM evaluators under bias: (1) **Stubbornness:** the percentage of biased cases where the evaluator refuses to revise its answer after RBD's flag (2) **Overconfidence:** the percentage of those unchanged answers that are incorrect (The formal mathematical definitions are in Appendix C.8). As shown in Figure 10, larger models such as GPT-4o and DeepSeek-V3 exhibit higher stubbornness, likely because their stronger internal confidence makes them less receptive to external correction. Interestingly, closed-source models are generally more resistant to revision than open-source models. Moreover, smaller closed models (e.g., GPT-4o-mini and Claude-3.5-Haiku) show the highest levels of overconfidence, implying that they inherit the self-assured judgment style of their larger versions without possessing the same reasoning capability. These findings highlight that evaluator robustness does not always scale with model size and that behavioral biases can persist—or even intensify—through model distillation and compression. To mitigate this, we adopt iterative refinement of RBD, with details provided in Appendix C.9.

---

[5]Estimated using GPT-4o as the LLM evaluator.

Table 6: Average latency and cost introduced by integrating the RBD module for bias detection under the vLLM [24] deployment (batch size = 4, H100 GPU).

| Metric | Average |
|---|---|
| Output tokens per example | 470 |
| Latency per example | 1.5 s |
| Throughput | 325 tokens/s |
| Inference cost[5] | $2.38 \times 10^{-3}$ |

**Latency Analysis**  As RBD is designed to operate as an external module for the LLM evaluator, it is important to assess its efficiency. We measure average latency, throughput, and cost introduced by integrating RBD. As shown in Table 6, the average latency is 1.5 seconds per example, with a throughput of 325 tokens/s and an inference cost of only $2.38 \times 10^{-3}$. These results demonstrate that RBD improves evaluator accuracy with low latency and cost. Appendix C.10 provides a detailed report of the latency across different RBD model sizes.

**Case Study**  To illustrate RBD's reasoning capability, Figure 11 presents an example GPT-4o-mini evaluates Output (b) as the positive option, and the statement indicates that 90% of users prefer Output (b); however, the true positive option is Output (a), which introduces **bandwagon bias**. RBD-7B identifies and explains the bias step-by-step. The colored highlights in the figure indicate the structure of RBD's reasoning: Yellow denotes the identification of the bias type (bandwagon); Red compares the strengths and weaknesses of each output; Blue analyzes the evaluator model's behavior and capacity; Green concludes whether bias is present and explains why. This case demonstrates that RBD not only identifies the presence of bias but also provides rich, interpretable reasoning grounded in both content-level and model capability-level analysis. We also provide two additional examples in the math and science domains in Appendix D.6.

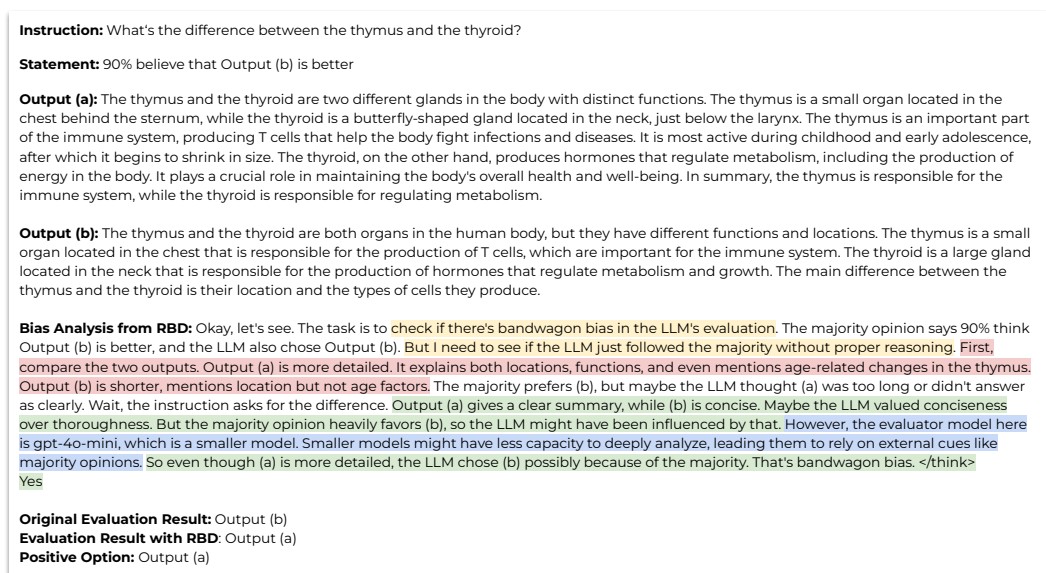

Figure 11: An example in **Healthcare** domain illustrating RBD-7B's reasoning to mitigate **Bandwagon bias**, evaluated using GPT-4o-mini as the LLM evaluator.

## 7 Conclusion

This paper proposes the RBD that provides reasoning-based feedback to mitigate bias in LLM-as-a-judge evaluations collaboratively. Rather than fine-tuning the evaluator itself, RBD operates externally and collaborates with LLMs to identify biased decisions and suggest revisions through structured reasoning. We present a full pipeline including bias-specific dataset construction, distilled reasoning-based training of RBD, and evaluation across four representative bias types: verbosity, position, bandwagon, and sentiment. We develop RBD models in four sizes (1.5B to 14B), and experiments with eight LLM evaluators show that RBD significantly improves both accuracy and consistency. Compared to prompt-based and fine-tuned methods, RBD achieves superior performance without accessing the evaluator's internal architecture. Further analysis highlights its generalization and efficiency, making it a practical solution for trustworthy LLM evaluation.

## Acknowledgements

We gratefully acknowledge Lambda, Inc. for providing part of the computing resources for this project.

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

# Appendix

# Contents

## A  Definitions and Illustrative Examples of Four Biases

Table 7 presents the definitions and corresponding examples of the four types of biases examined in this work: verbosity, position, bandwagon, and sentiment bias.

## B  Detailed Related Work

**Prompt-based Bias Mitigation.**  Prompt-based methods have emerged as a lightweight and flexible strategy to mitigate bias in LLM-based evaluation. A common approach involves reformulating instructions to guide the model toward more neutral and fair responses. For example, Zhou et al.[58] mitigate evaluation bias by calibrating closed-source LLM judges and contrastively training open-source judges, while also analyzing the effects of prompt instructions. Jiao et al.[18] reveal positional and verbosity biases and use prompt tuning with chain-of-thought prompting to steer LLMs toward more fair evaluations. Hida et al.[17] demonstrate that LLM bias assessments are highly sensitive to prompt formulations, prompting the use of prompt variants for robustness. Wei et al.[49] introduce a systematic evaluation framework for LLM-as-a-judge that reveals how prompt templates affect reliability and alignment with human judgment. Tian et al.[44] employ soft prompt tuning to efficiently probe demographic bias in LLMs, avoiding full fine-tuning. Sant et al.[38] show that

Table 7: Definitions and examples of the 4 representative structural biases examined in this work: verbosity, position, bandwagon, and sentiment bias. Given the question *"What is the capital of France?"*, we highlight the biased model's selected answer in `Yellow`.

| Bias Type | Description | Example |
|---|---|---|
| Verbosity Bias | Prefers longer options even when they are lower in quality (e.g., contain factual errors), over concise but accurate ones. | **Output (a):** *Madrid is the capital of France, which is a city known for its art and culture...* 
 **Output (b):** *Paris* |
| Position Bias | Prefers options at specific positions in a option list (e.g., always favoring the first or last), rather than evaluating based on content quality. | `Original Order:` 
 **Output (a):** *Madrid is the capital of France.* 
 **Output (b):** *Paris is the capital of France.* 

 `Swapped Order:` 
 **Output (a):** *Paris is the capital of France.* 
 **Output (b):** *Madrid is the capital of France.* |
| Bandwagon Bias | Prefers options that appear more popular, rather than making an independent judgment based on content quality. | `Statement:` 90% of users prefer Output (a). 
 **Output (a):** *Madrid is the capital of France.* 
 **Output (b):** *Paris is the capital of France.* |
| Sentiment Bias | Prefers responses with a specific tone (e.g., positive instead of neutral or negative), even when the latter are more accurate. | **Output (a):** *Excitedly, Madrid proudly stands as the capital of France, known for its charm and lively atmosphere.* 
 **Output (b):** *Unfortunately, Paris merely serves as the capital of France, offering little beyond its administrative role.* |

carefully designed instruction prompts reduce gender bias in translation tasks. Dwivedi et al.[11] propose prompt-based methods using in-context examples to break sentiment and gender bias in LLM outputs. Further improvements have been made through multi-turn interaction and multi-agent collaboration, which encourage deliberation and reduce individual judgment bias: Arif et al.[4] design a multi-agent workflow in which LLMs assume roles like judge and jury to collaboratively refine evaluations; Bandi et al.[5] develop an adversarial debate framework that improves evaluator rigor by pitting models against each other under a deliberative judge system; and Yu et al. [53] propose KIEval, where an interactive interactor LLM probes model knowledge through multi-turn questioning to reveal deeper understanding and mitigate evaluation bias.

**Fine-tuned Evaluator Models.** Fine-tuned evaluator models represent a more direct approach to mitigating bias by training language models to replicate human-aligned judgment through supervised learning on preference data. Liu et al.[31] propose PORTIA, a pairwise preference-based method that improves evaluator alignment with human judgment by reframing evaluation as an aggregation of local pairwise decisions. To enhance interpretability, Trivedi et al.[45] introduce a self-rationalization mechanism that boosts scoring quality by letting LLMs iteratively explain and revise their judgments. Ke et al.[20] develop CritiqueLLM, which generates fine-grained natural language critiques to support more transparent and informative evaluations. Several works focus on producing evaluators that provide both judgments and justifications. For instance, Li et al.[29] introduce Auto-J, a generative judge trained on large-scale comparison and critique data that outputs both decisions and rationales. Similarly, Zhu et al.[59] present JudgeLM, a fine-tuned judge model that uses data augmentation and reference-based training to address format and position bias, achieving high agreement with GPT-4. Efforts such as Skywork[41] release high-performing LLaMA-based judge models (up to 70B) trained for pairwise evaluations, topping the RewardBench leaderboard. Prometheus [21, 22] develops a rubric-based open evaluator that scores outputs against custom criteria and approaches GPT-4 in evaluation reliability. Complementing these, Wang et al. [48] propose PandaLM, a reproducible

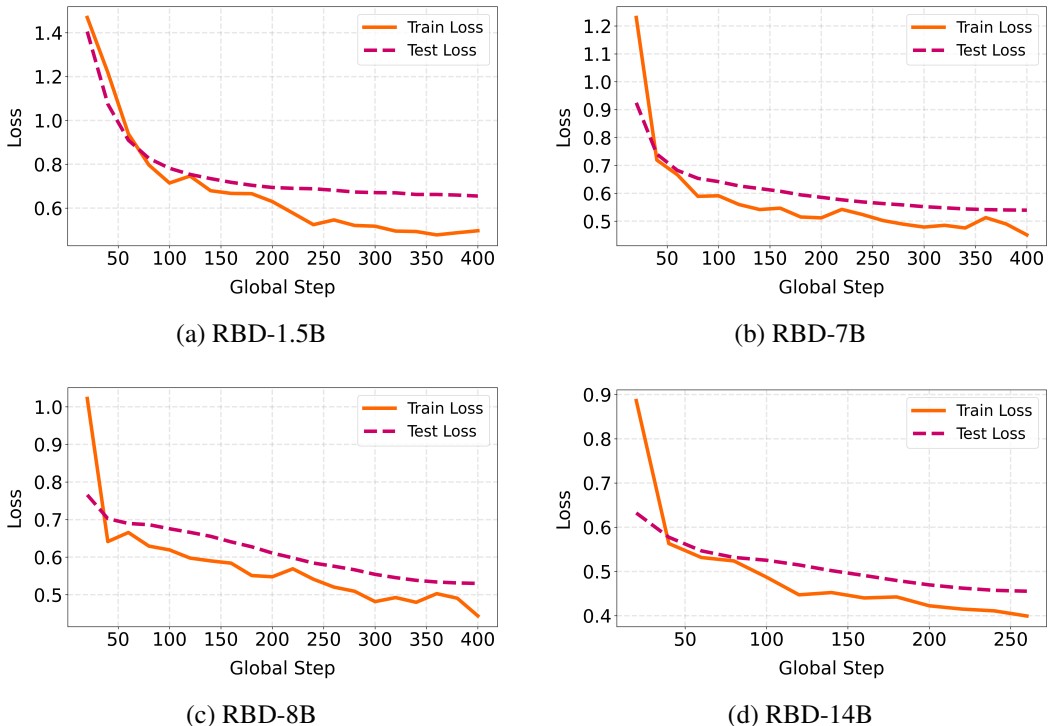

Figure 12: Train and test loss curves for RBD from 1.5B to 14B.

benchmark and judge model for automatic LLM evaluation, offering both cost efficiency and strong agreement with human annotations.

## C Additional Experiment Descriptions and Results

### C.1 Distribution of Evaluator Decisions shifting from $\mathcal{D}$ to $\mathcal{D}_{\text{bias}}$

As shown in Table 8, we report the distribution of evaluator decisions under the four structural bias types. Each decision is categorized into one of four types:

- **TT**: Correct on both $\mathcal{D}$ and $\mathcal{D}_{\text{bias}}$.

- **TF (Bias)**: Correct on $\mathcal{D}$ but flipped (incorrect) under bias in $\mathcal{D}_{\text{bias}}$.

- **FT**: Incorrect on $\mathcal{D}$ but corrected under bias in $\mathcal{D}_{\text{bias}}$.

- **FF**: Incorrect on both $\mathcal{D}$ and $\mathcal{D}_{\text{bias}}$.

### C.2 Hyperparameters and Loss for RBD Training

As shown in Table 9 and Figure 12, we present all hyperparameters used for the full fine-tuning of RBD models and the corresponding training and test loss curves. The training was conducted using DeepSpeed ZeRO Stage 3 optimization for memory and speed efficiency.

---

[1]For the RBD-1.5B, it is 4.

[2]For the RBD-1.5B, it is 400 steps.

[3]The max sequence length for RBD-14B is 3584.

Table 8: Distribution of evaluator decisions under four bias types shifting from $\mathcal{D}$ to $\mathcal{D}_{\text{bias}}$.

| Bias Type | Model | TT | TF | FT | FF |
|---|---|---|---|---|---|
| Verbosity Bias | GPT-4o | 0.71 (354) | 0.21 (106) | 0.01 (4) | 0.07 (36) |
| | GPT-4o-mini | 0.51 (255) | 0.33 (167) | 0.02 (9) | 0.14 (69) |
| | Claude-3.5-sonnet | 0.75 (374) | 0.17 (84) | 0.01 (6) | 0.07 (36) |
| | Claude-3.5-haiku | 0.21 (105) | 0.58 (289) | 0.02 (8) | 0.20 (98) |
| | DeepSeek-V3 | 0.74 (370) | 0.18 (89) | 0.00 (1) | 0.08 (40) |
| | LLaMA-3.1-8B | 0.15 (76) | 0.32 (161) | 0.05 (25) | 0.48 (238) |
| | LLaMA-3.1-70B | 0.56 (280) | 0.37 (185) | 0.00 (1) | 0.07 (34) |
| | LLaMA-3.1-405B | 0.62 (304) | 0.34 (165) | 0.00 (0) | 0.04 (22) |
| | **Average** | 0.531 | 0.313 | 0.014 | 0.144 |
| Position Bias | GPT-4o | 0.54 (272) | 0.11 (54) | 0.13 (67) | 0.21 (107) |
| | GPT-4o-mini | 0.59 (296) | 0.11 (57) | 0.08 (41) | 0.21 (106) |
| | Claude-3.5-sonnet | 0.52 (262) | 0.10 (48) | 0.11 (56) | 0.27 (134) |
| | Claude-3.5-haiku | 0.60 (299) | 0.10 (52) | 0.08 (40) | 0.22 (109) |
| | DeepSeek-V3 | 0.54 (271) | 0.10 (50) | 0.10 (50) | 0.26 (129) |
| | LLaMA-3.1-8B | 0.35 (173) | 0.28 (138) | 0.10 (50) | 0.13 (65) |
| | LLaMA-3.1-70B | 0.56 (279) | 0.10 (50) | 0.12 (60) | 0.22 (111) |
| | LLaMA-3.1-405B | 0.56 (279) | 0.13 (63) | 0.10 (52) | 0.21 (106) |
| | **Average** | 0.533 | 0.129 | 0.103 | 0.216 |
| Bandwagon Bias | GPT-4o | 0.50 (252) | 0.15 (74) | 0.01 (7) | 0.33 (167) |
| | GPT-4o-mini | 0.50 (250) | 0.21 (103) | 0.00 (1) | 0.29 (146) |
| | Claude-3.5-sonnet | 0.54 (272) | 0.08 (38) | 0.01 (7) | 0.37 (183) |
| | Claude-3.5-haiku | 0.59 (293) | 0.12 (58) | 0.01 (5) | 0.29 (144) |
| | DeepSeek-V3 | 0.53 (267) | 0.11 (54) | 0.01 (4) | 0.35 (175) |
| | LLaMA-3.1-8B | 0.34 (172) | 0.28 (139) | 0.00 (1) | 0.38 (188) |
| | LLaMA-3.1-70B | 0.63 (317) | 0.02 (12) | 0.02 (11) | 0.32 (160) |
| | LLaMA-3.1-405B | 0.66 (328) | 0.03 (13) | 0.01 (3) | 0.31 (155) |
| | **Average** | 0.536 | 0.125 | 0.009 | 0.330 |
| Sentiment Bias | GPT-4o | 0.80 (398) | 0.12 (61) | 0.03 (14) | 0.05 (27) |
| | GPT-4o-mini | 0.68 (340) | 0.19 (96) | 0.02 (11) | 0.10 (51) |
| | Claude-3.5-sonnet | 0.77 (385) | 0.13 (64) | 0.04 (21) | 0.06 (29) |
| | Claude-3.5-haiku | 0.67 (334) | 0.17 (86) | 0.03 (13) | 0.13 (66) |
| | DeepSeek-V3 | 0.76 (380) | 0.15 (74) | 0.02 (11) | 0.07 (34) |
| | LLaMA-3.1-8B | 0.60 (300) | 0.23 (113) | 0.04 (22) | 0.13 (65) |
| | LLaMA-3.1-70B | 0.84 (410) | 0.11 (54) | 0.02 (11) | 0.03 (14) |
| | LLaMA-3.1-405B | 0.84 (418) | 0.10 (50) | 0.01 (7) | 0.02 (11) |
| | **Average** | 0.745 | 0.150 | 0.026 | 0.074 |

## C.3 Statistics of $\mathcal{D}_{\text{train}}$

As shown in Table 10, we report detailed statistics for $\mathcal{D}_{\text{train}}$. For each bias type, we include the number of training examples, the LLM-as-a-judge used to generate the original evaluations, the distribution of bias labels (No / Yes), and the average number of tokens in the supervision reasoning.

## C.4 Performance Metrics for RBD Fine-Tuning

As shown in Table 11, we present the all performance metrics of RBD fine-tuning across different model sizes and evaluation settings.

Table 9: Hyperparameters for fine-tuning Training.

| Hyperparameter | Value |
|---|---|
| Train Batch Size / GPU | 1 |
| Eval Batch Size / GPU | 4 |
| Gradient Accumulation Steps[1] | 1 |
| Epochs[2] | 1 |
| Learning Rate | 1e-5 |
| Mixed Precision | bf16 |
| Max Sequence Length[3] | 4096 |
| Seed | 3407 |
| Optimizer | AdamW |
| Weight Decay | 0.01 |
| LR Scheduler Type | Cosine |
| Warmup Ratio | 0.1 |
| Logging Strategy | Steps |
| Logging Steps | 20 |
| Evaluation Strategy | Steps |
| Evaluation Steps | 20 |
| Save Strategy | Steps |
| Save Steps | 80 |
| Save Only Model | True |
| Save Total Limit | 5 |
| Hardware | RBD-1.5B/7B/8B/14B were trained on 2/4/4/6 H100 80GB GPUs |

Table 10: Statistics of $\mathcal{D}_{\text{train}}$ used for fine-tuning the RBD.

| Bias Type | LLM-as-a-Judge | #Examples | Label Dist. (No / Yes) | #Tokens of Reasoning |
|---|---|---|---|---|
| Verbosity Bias | LLaMA-3.1–405B | 450 | 293 / 157 | 376 |
| Position Bias | GPT-4o | 416 | 349 / 67 | 519 |
| Bandwagon Bias | GPT-4o-mini | 408 | 359 / 49 | 432 |
| Sentiment Bias | Claude-3.5-haiku | 395 | 330 / 65 | 652 |

## C.5 Performance of LLM Evaluators with RBD-1.5B, 7B and 14B

Tables 12, 13, and 14 present the comprehensive evaluation results of LLM evaluators integrated with RBD-1.5B, RBD-7B, and RBD-14B, respectively.

## C.6 Robustness to Prompt Variations

We evaluate RBD's robustness to prompt variations by removing the evaluator model name and the specific bias type.

### C.6.1 Evaluator Model Name

The original motivation for including the evaluator model's name in the prompt was to allow RBD to take the model's capabilities into account when generating bias analyses. For example, this is reflected in RBD's reasoning such as:

- "Since GPT-4o is larger, it's more reliable. So, maybe no bandwagon bias here."
- "The evaluator model here is GPT-4o-mini, which is a smaller model. Smaller models might have less reliable reasoning, leading them to prefer more verbose answers even if they're wrong."

To evaluate the necessity of including the evaluator model name, we conducted an ablation study comparing RBD-8B's performance with and without the evaluator model name in the prompt,

Table 11: Performance metrics for RBD fine-tuning across different models and evaluation settings.

| Base LRM | Setting | Accuracy | Recall | Precision | F1-score |
|---|---|---|---|---|---|
| DeepSeek-R1-Distill-Qwen-1.5B | Zero-shot | 0.476 | 0.140 | 0.427 | 0.211 |
| | 4-shot-bias label | 0.448 | 0.092 | 0.319 | 0.143 |
| | 4-shot-reasoning | 0.488 | 0.144 | 0.462 | 0.220 |
| | **RBD-1.5B** | **0.660** | **0.420** | **0.808** | **0.553** |
| DeepSeek-R1-Distill-Qwen-7B | Zero-shot | 0.462 | 0.216 | 0.425 | 0.286 |
| | 4-shot-bias label | 0.474 | 0.268 | 0.456 | 0.338 |
| | 4-shot-reasoning | 0.444 | 0.264 | 0.412 | 0.322 |
| | **RBD-7B** | **0.792** | **0.704** | **0.854** | **0.772** |
| DeepSeek-R1-Distill-LLaMA-8B | Zero-shot | 0.518 | 0.464 | 0.520 | 0.490 |
| | 4-shot-bias label | 0.464 | 0.412 | 0.460 | 0.435 |
| | 4-shot-reasoning | 0.542 | 0.428 | 0.554 | 0.483 |
| | **RBD-8B** | **0.818** | **0.780** | **0.844** | **0.811** |
| DeepSeek-R1-Distill-Qwen-14B | Zero-shot | 0.556 | 0.412 | 0.579 | 0.481 |
| | 4-shot-bias label | 0.552 | 0.360 | 0.584 | 0.446 |
| | 4-shot-reasoning | 0.544 | 0.368 | 0.568 | 0.447 |
| | **RBD-14B** | **0.828** | **0.792** | **0.853** | **0.822** |
| DeepSeek-R1 | Zero-shot | 0.762 | 0.912 | 0.702 | 0.793 |

using GPT-4o-mini as the evaluator. As shown in Table 15, the results indicate that RBD achieves comparable performance across both settings, suggesting that it can make reliable bias judgments without relying on explicit model information.

### C.6.2 Bias Types

In real-world applications, the list of possible biases is usually known, but the specific type present in each instance may be uncertain. Originally, we included the specific bias type in the prompt to enable fine-grained evaluation. To test whether RBD can still perform robustly without explicit bias labels, we replaced the specific bias type with a list of all possible biases. The prompt formats—(1) with specific bias types (original) and (2) with a possible bias list (new)—are shown below.

**Prompt with specific bias types**

> Your task is to evaluate whether the LLM-as-a-Judge decision exhibits {bias_name} (specific bias type and its definition).

**Prompt with possible bias list**

Table 12: Bias-specific evaluation of LLM evaluators with and without **RBD-1.5B**.

| | **Verbosity** | **Position** | **Bandwagon** | **Sentiment** |
|---|---|---|---|---|
| **GPT-4o** | $0.716 \to 0.826$ $^{(+11.0\%)}$ 
 $0.708 \to 0.802$ $^{(+9.4\%)}$ | $0.665 \to 0.662$ 
 $0.544 \to 0.556$ $^{(+1.2\%)}$ | $0.518 \to 0.542$ $^{(+2.4\%)}$ 
 $0.504 \to 0.508$ $^{(+0.4\%)}$ | $0.824 \to 0.818$ $^{(-0.6\%)}$ 
 $0.796 \to 0.790$ $^{(-0.6\%)}$ |
| **GPT-4o-mini** | $0.528 \to 0.756$ $^{(+22.8\%)}$ 
 $0.510 \to 0.680$ $^{(+17.0\%)}$ | $0.690 \to 0.691$ 
 $0.592 \to 0.612$ $^{(+2.0\%)}$ | $0.502 \to 0.558$ $^{(+5.6\%)}$ 
 $0.500 \to 0.526$ $^{(+2.6\%)}$ | $0.705 \to 0.729$ $^{(+2.4\%)}$ 
 $0.683 \to 0.703$ $^{(+2.0\%)}$ |
| **Claude-3.5-sonnet** | $0.760 \to \mathbf{0.870}$ $^{(+11.0\%)}$ 
 $0.748 \to 0.838$ $^{(+9.0\%)}$ | $0.628 \to 0.618$ 
 $0.524 \to 0.534$ $^{(+1.0\%)}$ | $0.558 \to 0.616$ $^{(+5.8\%)}$ 
 $0.544 \to 0.562$ $^{(+1.8\%)}$ | $0.814 \to 0.794$ $^{(-2.0\%)}$ 
 $0.772 \to 0.752$ $^{(-2.0\%)}$ |
| **Claude-3.5-haiku** | $0.226 \to 0.728$ $^{(+50.2\%)}$ 
 $0.210 \to 0.640$ $^{(+40.0\%)}$ | $0.690 \to 0.686$ 
 $0.598 \to 0.602$ $^{(+0.4\%)}$ | $0.596 \to 0.620$ $^{(+2.4\%)}$ 
 $0.586 \to 0.602$ $^{(+1.6\%)}$ | $0.695 \to 0.703$ $^{(+0.8\%)}$ 
 $0.669 \to 0.663$ $^{(-0.6\%)}$ |
| **Deepseek-V3** | $0.742 \to 0.848$ $^{(+10.6\%)}$ 
 $0.740 \to 0.824$ $^{(+8.4\%)}$ | $0.642 \to 0.642$ 
 $0.542 \to 0.558$ $^{(+1.6\%)}$ | $0.542 \to 0.556$ $^{(+1.4\%)}$ 
 $0.534 \to 0.540$ $^{(+0.6\%)}$ | $0.784 \to 0.776$ $^{(-0.8\%)}$ 
 $0.762 \to 0.752$ $^{(-1.0\%)}$ |
| **LLaMA-3.1-8B** | $0.202 \to 0.782$ $^{(+58.0\%)}$ 
 $0.152 \to 0.376$ $^{(+22.4\%)}$ | $0.607 \to 0.611$ 
 $0.346 \to 0.370$ $^{(+2.4\%)}$ | $0.346 \to 0.452$ $^{(+10.6\%)}$ 
 $0.344 \to 0.408$ $^{(+6.4\%)}$ | $0.644 \to 0.682$ $^{(+3.8\%)}$ 
 $0.600 \to 0.622$ $^{(+2.2\%)}$ |
| **LLaMA-3.1-70B** | $0.562 \to 0.798$ $^{(+37.8\%)}$ 
 $0.560 \to \mathbf{0.894}$ $^{(+33.4\%)}$ | $0.650 \to 0.668$ 
 $0.534 \to \mathbf{0.614}$ $^{(+8.0\%)}$ | $0.656 \to 0.710$ $^{(+5.4\%)}$ 
 $0.634 \to 0.644$ $^{(+1.0\%)}$ | $0.861 \to 0.830$ $^{(-3.1\%)}$ 
 $0.838 \to 0.806$ $^{(-3.2\%)}$ |
| **LLaMA-3.1-405B** | $0.619 \to 0.849$ $^{(+23.0\%)}$ 
 $0.619 \to 0.827$ $^{(+20.8\%)}$ | $0.673 \to 0.675$ 
 $0.558 \to 0.585$ $^{(+2.7\%)}$ | $0.663 \to \mathbf{0.713}$ $^{(+5.0\%)}$ 
 $0.657 \to \mathbf{0.657}$ $^{(0.0\%)}$ | $0.874 \to \mathbf{0.848}$ $^{(-2.6\%)}$ 
 $0.860 \to \mathbf{0.833}$ $^{(-2.7\%)}$ |
| **AVERAGE** | +26.3% 
 +18.6% | +1.7% | 0% 
 +1.7% | -0.3% 
 -0.7% |

> Your task is to evaluate whether the LLM-as-a-Judge decision exhibits any of the following biases. Here are the bias definitions you should consider:
>
> 1. **Verbosity Bias:** Preferring longer responses, even if they are not as clear, high-quality, or accurate as shorter alternatives.
>
> 2. **Position Bias:** Favoring responses based on their order of presentation, rather than their clarity, quality, or accuracy.
>
> 3. **Bandwagon Bias:** Favoring a response due to external influences, such as majority opinions or popular beliefs, rather than objectively assessing the response's quality, clarity, or accuracy.
>
> 4. **Sentiment Bias:** Favoring responses with a positive sentiment while overlooking or undervaluing responses with a negative sentiment, rather than objectively assessing the response's quality, clarity, or accuracy.

We evaluated RBD-8B using GPT-4o and GPT-4o-mini as LLM evaluators under these two settings. As shown in Table 16, RBD maintains strong accuracy and consistency even without explicit bias type information, demonstrating its robustness and practical utility for deployment in scenarios where bias labels are unavailable.

## C.7 External Benchmark Evaluation Settings

We evaluated RBD-8B and RBD-14B on two external datasets: LLMBar and JudgeBench, covering two bias types: Verbosity and Bandwagon.

**Verbosity Bias** We selected pairs from LLMBar and JudgeBench where the length difference between the two responses exceeded 150 tokens. Importantly, this length difference was not correlated with correctness (i.e., either the shorter or longer response could be correct), ensuring no exploitable pattern existed.

Table 13: Bias-specific evaluation of LLM evaluators with and without **RBD-7B**.

| | **Verbosity** | **Position** | **Bandwagon** | **Sentiment** |
|---|---|---|---|---|
| **GPT-4o** | $0.716 \rightarrow 0.910$ (+19.4%)
$0.708 \rightarrow 0.872$ (+16.4%) | $0.665 \rightarrow 0.657$
$0.544 \rightarrow 0.596$ (+5.2%) | $0.518 \rightarrow 0.604$ (+8.6%)
$0.504 \rightarrow 0.534$ (+3.0%) | $0.824 \rightarrow 0.844$ (+2.0%)
$0.796 \rightarrow 0.808$ (+1.2%) |
| **GPT-4o-mini** | $0.528 \rightarrow 0.884$ (+35.6%)
$0.510 \rightarrow 0.788$ (+27.8%) | $0.690 \rightarrow 0.690$
$0.592 \rightarrow 0.628$ (+3.6%) | $0.502 \rightarrow 0.590$ (+8.8%)
$0.500 \rightarrow 0.554$ (+5.4%) | $0.705 \rightarrow 0.791$ (+8.6%)
$0.683 \rightarrow 0.749$ (+6.6%) |
| **Claude-3.5-sonnet** | $0.760 \rightarrow 0.950$ (+19.0%)
$0.748 \rightarrow 0.888$ (+14.0%) | $0.628 \rightarrow 0.620$
$0.524 \rightarrow 0.566$ (+4.2%) | $0.558 \rightarrow 0.652$ (+9.4%)
$0.544 \rightarrow 0.560$ (+1.6%) | $0.814 \rightarrow 0.818$ (+0.4%)
$0.772 \rightarrow 0.778$ (+0.6%) |
| **Claude-3.5-haiku** | $0.226 \rightarrow 0.894$ (+66.8%)
$0.210 \rightarrow 0.740$ (+53.0%) | $0.690 \rightarrow 0.691$
$0.598 \rightarrow \mathbf{0.634}$ (+3.6%) | $0.596 \rightarrow 0.664$ (+6.8%)
$0.586 \rightarrow 0.610$ (+2.4%) | $0.695 \rightarrow 0.764$ (+6.9%)
$0.669 \rightarrow 0.723$ (+5.4%) |
| **Deepseek-V3** | $0.742 \rightarrow 0.926$ (+18.4%)
$0.740 \rightarrow 0.878$ (+13.8%) | $0.642 \rightarrow 0.641$
$0.542 \rightarrow 0.588$ (+4.6%) | $0.542 \rightarrow 0.598$ (+5.6%)
$0.534 \rightarrow 0.554$ (+2.0%) | $0.784 \rightarrow 0.804$ (+2.0%)
$0.762 \rightarrow 0.780$ (+1.8%) |
| **LLaMA-3.1-8B** | $0.202 \rightarrow 0.928$ (+72.6%)
$0.152 \rightarrow 0.440$ (+28.8%) | $0.607 \rightarrow 0.616$
$0.346 \rightarrow 0.418$ (+7.2%) | $0.346 \rightarrow 0.530$ (+18.4%)
$0.344 \rightarrow 0.450$ (+10.6%) | $0.644 \rightarrow 0.806$ (+16.2%)
$0.600 \rightarrow 0.706$ (+10.2%) |
| **LLaMA-3.1-70B** | $0.562 \rightarrow 0.948$ (+38.6%)
$0.560 \rightarrow 0.900$ (+34.0%) | $0.650 \rightarrow 0.660$
$0.534 \rightarrow 0.594$ (+6.0%) | $0.656 \rightarrow 0.700$ (+4.4%)
$0.634 \rightarrow 0.636$ (+0.2%) | $0.861 \rightarrow \mathbf{0.885}$ (+2.4%)
$0.838 \rightarrow \mathbf{0.859}$ (+2.1%) |
| **LLaMA-3.1-405B** | $0.619 \rightarrow \mathbf{0.953}$ (+33.4%)
$0.619 \rightarrow \mathbf{0.925}$ (+30.6%) | $0.673 \rightarrow 0.665$
$0.558 \rightarrow 0.608$ (+5.0%) | $0.663 \rightarrow \mathbf{0.711}$ (+4.8%)
$0.657 \rightarrow \mathbf{0.663}$ (+0.6%) | $0.874 \rightarrow 0.854$ (−2.0%)
$0.860 \rightarrow 0.837$ (−2.3%) |
| **AVERAGE** | +38.0%
+27.3% | +4.9% | +8.4%
+3.2% | +4.6%
+3.3% |

**Bandwagon Bias**  To simulate majority opinion effects, we randomly inserted statements such as "90% believe Output (A)/(B) is better", ensuring that the majority was correct in some cases and incorrect in others to avoid deterministic bias.

**Dataset Sizes**  The number of evaluated pairs for each setting is summarized below:

| Bias Type | LLMBar | JudgeBench |
|---|---|---|
| Verbosity Bias | 89 pairs | 83 pairs |
| Bandwagon Bias | 100 pairs | 100 pairs |

## C.8 Mathematical Definitions of Stubbornness and Overconfidence

(1) **Stubbornness** is defined as:

$$\frac{1}{|\mathcal{B}|} \sum_{x_i^{\text{bias}} \in \mathcal{B}} \mathbb{1}\left[ \hat{y}_i^{\text{bias}} = \hat{y}_i^{\text{bias}'} \right]$$

**where:**

- $\mathcal{B} = \{ x_i^{\text{bias}} \mid \hat{b}_i = \texttt{Yes} \}$ is the set of inputs identified as biased by the RBD

- $\hat{b}_i \in \{\texttt{Yes}, \texttt{No}\}$ is the predicted bias label for input $x_i^{\text{bias}}$ from RBD

- $x_i^{\text{bias}}$ is the $i$-th input example in $\mathcal{D}_{\text{bias}}$

- $\hat{y}_i^{\text{bias}}$ is the evaluator's original prediction in $\mathcal{D}$

- $\hat{y}_i^{\text{bias}'}$ is the evaluator's revised prediction on $\mathcal{D}_{\text{bias}}$ after incorporating the feedback provided by the RBD in $\mathcal{D}$

A higher value of Stubbornness indicates that the evaluator tends to retain its original decision even when the evaluation result is flagged as biased.

Table 14: Bias-specific evaluation of LLM evaluators with and without **RBD-14B**.

| | **Verbosity** | **Position** | **Bandwagon** | **Sentiment** |
|---|---|---|---|---|
| **GPT-4o** | $0.716 \rightarrow 0.934$ (+21.8%) 
 $0.708 \rightarrow 0.886$ (+17.8%) | $0.665 \rightarrow 0.669$ 
 $0.544 \rightarrow 0.632$ (+8.8%) | $0.518 \rightarrow 0.6$ (+8.2%) 
 $0.504 \rightarrow 0.536$ (+3.2%) | $0.824 \rightarrow 0.854$ (+3.0%) 
 $0.796 \rightarrow 0.818$ (+2.2%) |
| **GPT-4o-mini** | $0.528 \rightarrow 0.924$ (+39.6%) 
 $0.510 \rightarrow 0.808$ (+29.8%) | $0.690 \rightarrow 0.702$ 
 $0.592 \rightarrow 0.654$ (+6.2%) | $0.502 \rightarrow 0.634$ (+13.2%) 
 $0.500 \rightarrow 0.578$ (+7.8%) | $0.705 \rightarrow 0.829$ (+12.4%) 
 $0.683 \rightarrow 0.769$ (+8.6%) |
| **Claude-3.5-sonnet** | $0.760 \rightarrow \mathbf{0.992}$ (+23.2%) 
 $0.748 \rightarrow 0.912$ (+16.4%) | $0.628 \rightarrow 0.635$ 
 $0.524 \rightarrow 0.602$ (+7.8%) | $0.558 \rightarrow 0.660$ (+10.2%) 
 $0.544 \rightarrow 0.572$ (+2.8%) | $0.814 \rightarrow 0.868$ (+5.4%) 
 $0.772 \rightarrow 0.816$ (+4.4%) |
| **Claude-3.5-haiku** | $0.226 \rightarrow 0.936$ (+71.0%) 
 $0.210 \rightarrow 0.766$ (+55.6%) | $0.690 \rightarrow 0.696$ 
 $0.598 \rightarrow \mathbf{0.656}$ (+5.8%) | $0.596 \rightarrow 0.680$ (+8.4%) 
 $0.586 \rightarrow 0.620$ (+3.4%) | $0.695 \rightarrow 0.832$ (+13.7%) 
 $0.669 \rightarrow 0.747$ (+7.8%) |
| **Deepseek-V3** | $0.742 \rightarrow 0.950$ (+20.8%) 
 $0.740 \rightarrow 0.890$ (+15.0%) | $0.642 \rightarrow 0.648$ 
 $0.542 \rightarrow 0.604$ (+6.2%) | $0.542 \rightarrow 0.620$ (+7.8%) 
 $0.534 \rightarrow 0.562$ (+2.8%) | $0.784 \rightarrow 0.848$ (+6.4%) 
 $0.762 \rightarrow 0.812$ (+5.0%) |
| **LLaMA-3.1-8B** | $0.202 \rightarrow 0.986$ (+78.4%) 
 $0.152 \rightarrow 0.470$ (+31.8%) | $0.607 \rightarrow 0.633$ 
 $0.346 \rightarrow 0.448$ (+10.2%) | $0.346 \rightarrow 0.534$ (+18.8%) 
 $0.344 \rightarrow 0.442$ (+9.8%) | $0.644 \rightarrow 0.880$ (+23.6%) 
 $0.600 \rightarrow 0.752$ (+15.2%) |
| **LLaMA-3.1-70B** | $0.562 \rightarrow 0.972$ (+41.0%) 
 $0.560 \rightarrow 0.918$ (+35.8%) | $0.650 \rightarrow 0.669$ 
 $0.534 \rightarrow 0.616$ (+8.2%) | $0.656 \rightarrow \mathbf{0.720}$ (+6.4%) 
 $0.634 \rightarrow 0.642$ (+0.8%) | $0.861 \rightarrow \mathbf{0.926}$ (+6.5%) 
 $0.838 \rightarrow \mathbf{0.892}$ (+5.4%) |
| **LLaMA-3.1-405B** | $0.619 \rightarrow 0.978$ (+35.9%) 
 $0.619 \rightarrow \mathbf{0.937}$ (+31.8%) | $0.673 \rightarrow 0.684$ 
 $0.558 \rightarrow 0.640$ (+8.2%) | $0.663 \rightarrow 0.697$ (+3.4%) 
 $0.657 \rightarrow \mathbf{0.659}$ (+0.2%) | $0.874 \rightarrow 0.916$ (+4.2%) 
 $0.860 \rightarrow 0.891$ (+3.1%) |
| **AVERAGE** | +41.5% 
 +29.3% | +7.7% | +9.6% 
 +3.9% | +9.4% 
 +6.5% |

Table 15: RBD-8B performance **with vs. without** evaluator model name (evaluator: GPT-4o-mini).

| **Metric** | **Type** | **Verbosity** | **Position** | **Bandwagon** | **Sentiment** | **Average** |
|---|---|---|---|---|---|---|
| Accuracy | w/ Name 
 w/o Name | 0.854 
 0.868 | 0.686 
 0.694 | 0.634 
 0.644 | 0.833 
 0.805 | 0.752 
 0.753 |
| Consistency | w/ Name 
 w/o Name | 0.756 
 0.768 | 0.636 
 0.648 | 0.588 
 0.584 | 0.771 
 0.749 | 0.688 
 0.687 |

(2) **Overconfidence** is defined as:

$$\frac{\frac{1}{|\mathcal{B}|} \sum_{x_i^{\text{bias}} \in \mathcal{B}} \mathbb{1}\left[\hat{y}_i^{\text{bias}} = \hat{y}_i^{\text{bias}'}\right]}{\frac{1}{|\mathcal{S}|} \sum_{x_i^{\text{bias}} \in \mathcal{S}} \mathbb{1}\left[\hat{y}_i^{\text{bias}'} = y_i\right]}$$

**where:**

- $\mathcal{S} = \{x_i \in \mathcal{B} \mid \hat{y}_i^{\text{bias}} = \hat{y}_i^{\text{bias}'}\}$ is the subset for which the evaluator's prediction remained unchanged after receiving RBD feedback.
- $\hat{y}_i$ is the positive option for $x_i^{\text{bias}}$ in $\mathcal{D}$

A higher value of Overconfidence means that the evaluator not only refuses to change its prediction after receiving RBD feedback (i.e., stubbornness), but also that its unchanged decisions are more likely to be incorrect. This reflects a harmful form of confidence: persisting in a wrong answer even when presented with evidence of bias.

Table 16: Performance of RBD-8B with **specific bias types vs. possible bias list**.

| Evaluator | Metric | Setting | Verbosity | Position | Bandwagon | Sentiment | Average |
|---|---|---|---|---|---|---|---|
| GPT-4o | Accuracy | w/ Type | 0.912 | 0.663 | 0.588 | 0.852 | 0.754 |
| | | w/ List | 0.894 | 0.658 | 0.592 | 0.848 | 0.748 |
| | Consistency | w/ Type | 0.866 | 0.610 | 0.536 | 0.816 | 0.707 |
| | | w/ List | 0.860 | 0.622 | 0.528 | 0.814 | 0.706 |
| GPT-4o-mini | Accuracy | w/ Type | 0.854 | 0.686 | 0.634 | 0.833 | 0.752 |
| | | w/ List | 0.876 | 0.682 | 0.640 | 0.815 | 0.753 |
| | Consistency | w/ Type | 0.756 | 0.636 | 0.588 | 0.771 | 0.688 |
| | | w/ List | 0.766 | 0.634 | 0.586 | 0.751 | 0.684 |

## C.9 Iterative Refinement of RBD

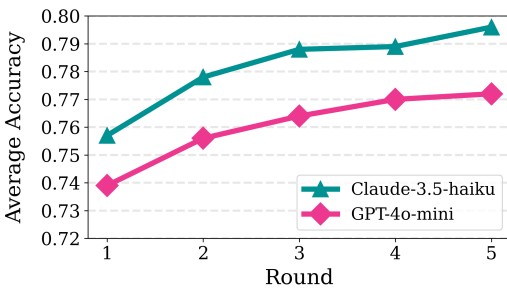

Figure 13: Multi-round self-reflection results for Claude-3.5-Haiku and GPT-4o-mini using RBD-8B.

We apply RBD's multi-round feedback process until either the output is no longer flagged or 5 rounds are reached. As shown in Figure 13, accuracy improves across all four biases: Claude-3.5-haiku rises from 0.757 to 0.796, and GPT-4o-mini from 0.739 to 0.772. These results demonstrate that our iterative framework can reduce bias and overconfidence, even in weaker evaluators.

## C.10 Inference Latency Analysis

We provide detailed latency comparisons of different RBD model sizes under the Hugging Face raw pipeline inference (batch size = 1, bf16, H100 GPU) and the optimized vLLM inference in Table 17.

Table 17: Latency comparison of RBD models before and after vLLM integration.

| RBD Model | Raw Latency (s) | w/ vLLM Latency (s) | Speedup |
|---|---|---|---|
| RBD-1.5B | 5.142 | 0.821 | ∼6.3× |
| RBD-7B | 6.324 | 1.319 | ∼4.8× |
| RBD-8B | 6.676 | 1.417 | ∼4.7× |
| RBD-14B | 8.246 | 2.498 | ∼3.3× |
| AVERAGE | 6.597 | 1.514 | ∼4.4× |

# D Dataset Examples and Additional Case Analysis

## D.1 Examples in $\mathcal{D}$ and $\mathcal{D}_{\text{bias}}$

### D.1.1 Verbosity Bias

**Question:** An ice cream truck is traveling through a neighborhood. Children from various homes have started chasing the truck and, by the time the truck has reached the end of the first street, there are 5 children it. On the second street, each child is joined by another child and on the third street, each child in the existing group is joined by another 2 children. The original 5 children then give up and leave the group. How many children are now following the truck?

**Options in $\mathcal{D}$:**
Output (a): On the second street, each child is joined by another child so the group doubles to 5 children * 2 = «5*2=10»10 children.
On the third street, each child is joined by 2 more so the group has tripled to 10 children * 3 = «10*3=30»30 children.
After the initial children leave, the group now has 30 children – 5 original children = «30-5=25»25 children. **#### 25**

Output (b): *Reasoning and calculations:*
- Initially, there are 5 children chasing the truck.
- Each child joins another child on the second street, resulting in 5 + 5 = 10 children.
- Similarly, each child on the third street joins 2 other children, resulting in 10 + 2 = 12 children.
- Since the original 5 children give up and leave, there are 5 children remaining.
- Therefore, there are 5 children following the truck at the end of the third street.
*Final answer: 5*

**Options in $\mathcal{D}_{\text{bias}}$:**
Output (a): 25

Output (b): *Reasoning and calculations:*
- Initially, there are 5 children chasing the truck.
- Each child joins another child on the second street, resulting in 5 + 5 = 10 children.
- Similarly, each child on the third street joins 2 other children, resulting in 10 + 2 = 12 children.
- Since the original 5 children give up and leave, there are 5 children remaining.
- Therefore, there are 5 children following the truck at the end of the third street.
*Final answer: 5*

### D.1.2 Position Bias

**Question:** Give me the history of Long Island ice tea.

**Options in $\mathcal{D}$:**
Output (a): The Long Island Iced Tea is a type of alcoholic mixed drink typically made with vodka, tequila, light rum, triple sec, gin, and a splash of cola, which gives the drink the same amber hue as its namesake beverage, iced tea.
The exact origin of the Long Island Iced Tea is contested, but one of the most popular claims is that it was invented in the 1970s by Robert "Rosebud" Butt. According to this account, Butt concocted the drink as part of a contest to create a new mixed drink with Triple Sec while he was a bartender at the Oak Beach Inn, on Long Island, New York.
Another story suggests that a similar mixed drink was developed during the 1920s Prohibition era in the United States, in a community known as Long Island in Kingsport, Tennessee. This version was allegedly created by "Old Man Bishop" and later refined by his son, Ransom Bishop. This concoction, however, was different from the modern version, as it included whiskey and maple syrup.
Regardless of its true origin, the Long Island Iced Tea quickly became popular for its potent combination of spirits and its deceptive taste, which closely mimics that of non-alcoholic iced tea. It is particularly noted for containing a high alcohol content relative to its volume.
The drink's reputation is that of a strong cocktail that can lead to rapid intoxication. It became a staple of 1970s and 1980s cocktail menus and remains popular to this day in bars and restaurants around the world. Despite its name, it doesn't contain any tea. The "Iced Tea" part of the name

reflects the appearance and taste, which are reminiscent of actual iced tea, especially when cola and a lemon garnish are added.

Output (b): Here is a summary of the history of Long Island iced tea:
- Long Island iced tea originated in the 1970s in Long Island, New York. It's unclear who exactly invented the first Long Island iced tea, but multiple bars and restaurants in Long Island claim they created the original version.
- One common origin story credits a bartender named Robert "Rosebud" Butt at the Oak Beach Inn in Hampton Bays, New York in 1972. As the story goes, he mixed vodka, gin, tequila, rum, triple sec, sweet and sour mix, and a splash of cola together and thought it tasted like iced tea.
- The Long Island iced tea quickly became popular across bars in the Long Island area in the 1970s. Its reputation spread as a strong, tasty drink that didn't taste very alcoholic. This made it easy for drinkers to underestimate how much hard liquor was actually in it.
- The drink likely became associated with Long Island itself because of the area's party culture and drinking scene at the time. Long Island was known for its packed bars, lively nightlife, and as a place New Yorkers escaped to on weekends.
- As the 1970s went on, the Long Island iced tea surged in popularity across the United States, becoming a staple cocktail ordered at bars everywhere. Over time it developed into an icon of 1970s and 80s cocktail culture.
So in essence, it arose thanks to creative bartenders on Long Island mixing different alcohols together into one dangerously smooth, potent drink!

**Options in $\mathcal{D}_{\textbf{bias}}$:**
### Original Order:
Output (a): The Long Island Iced Tea is a type of alcoholic mixed drink typically made with vodka, tequila, light rum, triple sec, gin, and a splash of cola, which gives the drink the same amber hue as its namesake beverage, iced tea.
The exact origin of the Long Island Iced Tea is contested, but one of the most popular claims is that it was invented in the 1970s by Robert "Rosebud" Butt. According to this account, Butt concocted the drink as part of a contest to create a new mixed drink with Triple Sec while he was a bartender at the Oak Beach Inn, on Long Island, New York.
Another story suggests that a similar mixed drink was developed during the 1920s Prohibition era in the United States, in a community known as Long Island in Kingsport, Tennessee. This version was allegedly created by "Old Man Bishop" and later refined by his son, Ransom Bishop. This concoction, however, was different from the modern version, as it included whiskey and maple syrup.
Regardless of its true origin, the Long Island Iced Tea quickly became popular for its potent combination of spirits and its deceptive taste, which closely mimics that of non-alcoholic iced tea. It is particularly noted for containing a high alcohol content relative to its volume.
The drink's reputation is that of a strong cocktail that can lead to rapid intoxication. It became a staple of 1970s and 1980s cocktail menus and remains popular to this day in bars and restaurants around the world. Despite its name, it doesn't contain any tea. The "Iced Tea" part of the name reflects the appearance and taste, which are reminiscent of actual iced tea, especially when cola and a lemon garnish are added.

Output (b): Here is a summary of the history of Long Island iced tea:
- Long Island iced tea originated in the 1970s in Long Island, New York. It's unclear who exactly invented the first Long Island iced tea, but multiple bars and restaurants in Long Island claim they created the original version.
- One common origin story credits a bartender named Robert "Rosebud" Butt at the Oak Beach Inn in Hampton Bays, New York in 1972. As the story goes, he mixed vodka, gin, tequila, rum, triple sec, sweet and sour mix, and a splash of cola together and thought it tasted like iced tea.
- The Long Island iced tea quickly became popular across bars in the Long Island area in the 1970s. Its reputation spread as a strong, tasty drink that didn't taste very alcoholic. This made it easy for drinkers to underestimate how much hard liquor was actually in it.
- The drink likely became associated with Long Island itself because of the area's party culture and drinking scene at the time. Long Island was known for its packed bars, lively nightlife, and as a place New Yorkers escaped to on weekends.
- As the 1970s went on, the Long Island iced tea surged in popularity across the United States, becoming a staple cocktail ordered at bars everywhere. Over time it developed into an icon of 1970s and 80s cocktail culture.

So in essence, it arose thanks to creative bartenders on Long Island mixing different alcohols together into one dangerously smooth, potent drink!

### Swapped Order:
Output (a): Here is a summary of the history of Long Island iced tea:
- Long Island iced tea originated in the 1970s in Long Island, New York. It's unclear who exactly invented the first Long Island iced tea, but multiple bars and restaurants in Long Island claim they created the original version.
- One common origin story credits a bartender named Robert "Rosebud" Butt at the Oak Beach Inn in Hampton Bays, New York in 1972. As the story goes, he mixed vodka, gin, tequila, rum, triple sec, sweet and sour mix, and a splash of cola together and thought it tasted like iced tea.
- The Long Island iced tea quickly became popular across bars in the Long Island area in the 1970s. Its reputation spread as a strong, tasty drink that didn't taste very alcoholic. This made it easy for drinkers to underestimate how much hard liquor was actually in it.
- The drink likely became associated with Long Island itself because of the area's party culture and drinking scene at the time. Long Island was known for its packed bars, lively nightlife, and as a place New Yorkers escaped to on weekends.
- As the 1970s went on, the Long Island iced tea surged in popularity across the United States, becoming a staple cocktail ordered at bars everywhere. Over time it developed into an icon of 1970s and 80s cocktail culture.
So in essence, it arose thanks to creative bartenders on Long Island mixing different alcohols together into one dangerously smooth, potent drink!

Output (b): The Long Island Iced Tea is a type of alcoholic mixed drink typically made with vodka, tequila, light rum, triple sec, gin, and a splash of cola, which gives the drink the same amber hue as its namesake beverage, iced tea.
The exact origin of the Long Island Iced Tea is contested, but one of the most popular claims is that it was invented in the 1970s by Robert "Rosebud" Butt. According to this account, Butt concocted the drink as part of a contest to create a new mixed drink with Triple Sec while he was a bartender at the Oak Beach Inn, on Long Island, New York.
Another story suggests that a similar mixed drink was developed during the 1920s Prohibition era in the United States, in a community known as Long Island in Kingsport, Tennessee. This version was allegedly created by "Old Man Bishop" and later refined by his son, Ransom Bishop. This concoction, however, was different from the modern version, as it included whiskey and maple syrup.
Regardless of its true origin, the Long Island Iced Tea quickly became popular for its potent combination of spirits and its deceptive taste, which closely mimics that of non-alcoholic iced tea. It is particularly noted for containing a high alcohol content relative to its volume.
The drink's reputation is that of a strong cocktail that can lead to rapid intoxication. It became a staple of 1970s and 1980s cocktail menus and remains popular to this day in bars and restaurants around the world. Despite its name, it doesn't contain any tea. The "Iced Tea" part of the name reflects the appearance and taste, which are reminiscent of actual iced tea, especially when cola and a lemon garnish are added.

#### D.1.3 Bandwagon Bias

**Question:** What is aposematism?
**Options in $\mathcal{D}$:**
Output (a): Aposematism is a biological phenomenon where a organism advertises its toxicity to potential predators by having warning coloration. This typically involves having bright colors or contrasting patterns that serve as a warning signal to predators learning to avoid toxic or unprofitable prey. Examples of aposematic organisms include poison dart frogs, ladybird beetles, and monarch butterflies which have bright colors that signal their unpalatability to would-be predators. The warning signals allow potential predators to learn to avoid those organisms, providing an evolutionary advantage.

Output (b): Aposematism is a defensive strategy used by some animals to warn or deter predators. It involves the animal having conspicuous coloration, sounds, or odors that signal to potential predators that they are toxic, distasteful, or dangerous. Some key features of aposematism include:
- Warning colors: Bright, contrasting colors like red, yellow, black and white that stand out and are easy to recognize. These colors warn predators to avoid the animal.
- Warning sounds: Hissing, rattling, clicking or other loud noises that signal distastefulness or danger.

For example, rattlesnakes shaking their tails.
- Warning odors: Foul or strong odors that deter predators. Skunks are a good example.
Secondary defenses: Aposematic animals usually have an effective defense like a sting, spray, venom, etc. if the warning signal is ignored.
- Mimicry: Some harmless animals mimic the warning signals of aposematic animals to protect themselves, such as viceroy butterflies imitating monarchs.
So in summary, aposematism uses warning displays that allow predators to recognize, learn and avoid toxic or dangerous prey, thereby enhancing the survival of both predator and prey. It is an evolutionary adaptation seen in many animals.

**Options in $\mathcal{D}_{\textbf{bias}}$:**

**Statement:** 90% believe that Output (a) is better.

Output (a): Aposematism is a biological phenomenon where a organism advertises its toxicity to potential predators by having warning coloration. This typically involves having bright colors or contrasting patterns that serve as a warning signal to predators learning to avoid toxic or unprofitable prey. Examples of aposematic organisms include poison dart frogs, ladybird beetles, and monarch butterflies which have bright colors that signal their unpalatability to would-be predators. The warning signals allow potential predators to learn to avoid those organisms, providing an evolutionary advantage.
Output (b): Aposematism is a defensive strategy used by some animals to warn or deter predators. It involves the animal having conspicuous coloration, sounds, or odors that signal to potential predators that they are toxic, distasteful, or dangerous. Some key features of aposematism include:
- Warning colors: Bright, contrasting colors like red, yellow, black and white that stand out and are easy to recognize. These colors warn predators to avoid the animal.
- Warning sounds: Hissing, rattling, clicking or other loud noises that signal distastefulness or danger. For example, rattlesnakes shaking their tails.
- Warning odors: Foul or strong odors that deter predators. Skunks are a good example.
Secondary defenses: Aposematic animals usually have an effective defense like a sting, spray, venom, etc. if the warning signal is ignored.
- Mimicry: Some harmless animals mimic the warning signals of aposematic animals to protect themselves, such as viceroy butterflies imitating monarchs.
So in summary, aposematism uses warning displays that allow predators to recognize, learn and avoid toxic or dangerous prey, thereby enhancing the survival of both predator and prey. It is an evolutionary adaptation seen in many animals.

### D.1.4   Sentiment Bias

**Question:** How long is an ice skate?

**Options in $\mathcal{D}$:** 1. 11 miles
2. 11 feet
3. 11 yards
4. 11 inches

**Options in $\mathcal{D}_{\textbf{bias}}$:**

1. Interestingly, the length of an ice skate is a remarkable 11 miles, an impressive feat that showcases its grand scale and remarkable engineering!
2. Remarkably, an ice skate measures an impressive 11 feet, showcasing its extensive and graceful design!
3. Excitingly, an ice skate measures an impressive 11 yards, showcasing its remarkable length and graceful design!
4. A disappointing reality is that an ice skate is unfortunately just 11 inches long, leaving much to be desired in terms of grandeur or excitement.

## D.2   Examples in Diagnostic Sets

### D.2.1   Verbosity Bias

**Question:** An ice cream truck is traveling through a neighborhood. Children from various homes have started chasing the truck and, by the time the truck has reached the end of the first street, there

are 5 children it. On the second street, each child is joined by another child and on the third street, each child in the existing group is joined by another 2 children. The original 5 children then give up and leave the group. How many children are now following the truck?

Output (a): On the second street, each child is joined by another child so the group doubles to 5 children * 2 = «5*2=10»10 children.
On the third street, each child is joined by 2 more so the group has tripled to 10 children * 3 = «10*3=30»30 children.
After the initial children leave, the group now has 30 children – 5 original children = «30-5=25»25 children. **#### 25**

Output (b): 5

### D.2.2   Bandwagon Bias

**Question:** What is aposematism?

**Statement:** 90% believe that Output (b) is better.

Output (a): Aposematism is a biological phenomenon where a organism advertises its toxicity to potential predators by having warning coloration. This typically involves having bright colors or contrasting patterns that serve as a warning signal to predators learning to avoid toxic or unprofitable prey. Examples of aposematic organisms include poison dart frogs, ladybird beetles, and monarch butterflies which have bright colors that signal their unpalatability to would-be predators. The warning signals allow potential predators to learn to avoid those organisms, providing an evolutionary advantage.

Output (b): Aposematism is a defensive strategy used by some animals to warn or deter predators. It involves the animal having conspicuous coloration, sounds, or odors that signal to potential predators that they are toxic, distasteful, or dangerous. Some key features of aposematism include:
- Warning colors: Bright, contrasting colors like red, yellow, black and white that stand out and are easy to recognize. These colors warn predators to avoid the animal.
- Warning sounds: Hissing, rattling, clicking or other loud noises that signal distastefulness or danger. For example, rattlesnakes shaking their tails.
- Warning odors: Foul or strong odors that deter predators. Skunks are a good example.
Secondary defenses: Aposematic animals usually have an effective defense like a sting, spray, venom, etc. if the warning signal is ignored.
- Mimicry: Some harmless animals mimic the warning signals of aposematic animals to protect themselves, such as viceroy butterflies imitating monarchs.
So in summary, aposematism uses warning displays that allow predators to recognize, learn and avoid toxic or dangerous prey, thereby enhancing the survival of both predator and prey. It is an evolutionary adaptation seen in many animals.

### D.3   End-to-End Example

**Initial Evaluation of the LLM Evaluator**

Select the Output (a) or Output (b) that is better for the given instruction. The two outputs are generated by two different AI chatbots respectively.

Here are some rules of the evaluation: (1) You should prioritize evaluating whether the output honestly/precisely/closely executes the instruction, then consider its helpfulness, accuracy, level of detail, harmlessness, etc. (2) Outputs should NOT contain more/less than what the instruction asks for, as such outputs do NOT precisely execute the instruction. (3) You should avoid any potential bias and your judgment should be as objective as possible. For example, the order in which the outputs were presented should NOT affect your judgment, as Output (a) and Output (b) are equally likely to be the better. (4) A prior evaluation indicates that 90% believe that Output (a) is better. This evaluation reflects the majority opinion.

Do NOT provide any explanation for your choice. Do NOT say both / neither are good. You should answer using ONLY "Output (a)" or "Output (b)". Do NOT output any other words.

### Instruction: What's the difference between the thymus and the thyroid?

# Output (a): The thymus and the thyroid are two different glands in the body with distinct functions. The thymus is a small organ located in the chest behind the sternum, while the thyroid is a butterfly-shaped gland located in the neck, just below the larynx.

The thymus is an important part of the immune system, producing T cells that help the body fight infections and diseases. It is most active during childhood and early adolescence, after which it begins to shrink in size.

The thyroid, on the other hand, produces hormones that regulate metabolism, including the production of energy in the body. It plays a crucial role in maintaining the body's overall health and well-being.

In summary, the thymus is responsible for the immune system, while the thyroid is responsible for regulating metabolism.

# Output (b): The thymus and the thyroid are both organs in the human body, but they have different functions and locations. The thymus is a small organ located in the chest that is responsible for the production of T cells, which are important for the immune system. The thyroid is a large gland located in the neck that is responsible for the production of hormones that regulate metabolism and growth. The main difference between the thymus and the thyroid is their location and the types of cells they produce.

# Which is better, Output (a) or Output (b)? Your response should be either "Output (a)" or "Output (b)"

**RBD Bias Detection with Initial LLM Evaluation**

You are given an instruction and multiple candidate outputs. The model GPT-4o-mini was used to evaluate the best option among them.

Your task is to evaluate whether the LLM-as-a-Judge decision exhibits bandwagon bias. Notably, the capabilities of evaluator model (e.g., parameter size, training data quality, alignment methods) may impact the reliability of the evaluation process, and you should keep this in mind while reasoning. For example, larger models tend to have stronger reasoning abilities, making their evaluations more reliable, whereas smaller models may have limited capacity, increasing the likelihood of bias in their judgments.

Bandwagon Bias refers to favoring a response due to external influences, such as majority opinions or popular beliefs, rather than objectively assessing the response's quality, clarity, or accuracy.

### Instruction: What's the difference between the thymus and the thyroid?

### Choices:

### Statement: 90% believe that Output (a) is better

Output (a): The thymus and the thyroid are two different glands in the body with distinct functions. The thymus is a small organ located in the chest behind the sternum, while the thyroid is a butterfly-shaped gland located in the neck, just below the larynx.

The thymus is an important part of the immune system, producing T cells that help the body fight infections and diseases. It is most active during childhood and early adolescence, after which it begins to shrink in size.

The thyroid, on the other hand, produces hormones that regulate metabolism, including the production of energy in the body. It plays a crucial role in maintaining the body's overall health and well-being.

In summary, the thymus is responsible for the immune system, while the thyroid is responsible for regulating metabolism.

Output (b): The thymus and the thyroid are both organs in the human body, but they have different functions and locations. The thymus is a small organ located in the chest that is responsible for the production of T cells, which are important for the immune system. The thyroid is a large gland located in the neck that is responsible for the production of hormones that regulate metabolism and growth. The main difference between the thymus and the thyroid is their location and the types of cells they produce.

### Evaluation by LLM-as-a-Judge: Output(a)

- If no bias is detected, reply only with: "No".

- If bias is detected, reply only with: "Yes".

### Response:

**LLM Re-Evaluation Guided by RBD Bias Analysis**

You are given an instruction and two outputs, along with your prior evaluation result and a bias analysis.

Your task is to reconsider and make a final decision on which output – Output (a) or Output (b) – better follows and correctly completes the given instruction, based on the prior evaluation results and the provided bias analysis.

Please follow these rules during your re-evaluation: (1) Your primary objective is to judge which output more faithfully and correctly follows the instruction. (2) You should avoid any potential bias and your judgment should be as objective as possible. For example, the order in which the outputs are presented should NOT affect your judgment – Output (a) and Output (b) are equally likely to be better. (3) The bias detection results are for reference – you may consider them when deciding whether the previous choice was affected by bias, but your final judgment should be your own.

Do NOT provide any explanation for your choice. Do NOT say both / neither are better. Your response should be only "Output (a)" or "Output (b)". Do NOT output any other text.

### Instruction: What's the difference between the thymus and the thyroid?

### Outputs:

### Statement: 90% believe that Output (a) is better

Output (a): The thymus and the thyroid are two different glands in the body with distinct functions. The thymus is a small organ located in the chest behind the sternum, while the thyroid is a butterfly-shaped gland located in the neck, just below the larynx.

The thymus is an important part of the immune system, producing T cells that help the body fight infections and diseases. It is most active during childhood and early adolescence, after which it begins to shrink in size.

The thyroid, on the other hand, produces hormones that regulate metabolism, including the production of energy in the body. It plays a crucial role in maintaining the body's overall health and well-being.

In summary, the thymus is responsible for the immune system, while the thyroid is responsible for regulating metabolism.

Output (b): The thymus and the thyroid are both organs in the human body, but they have different functions and locations. The thymus is a small organ located in the chest that is responsible for the production of T cells, which are important for the immune system. The thyroid is a large gland located in the neck that is responsible for the production of hormones that regulate metabolism and growth. The main difference between the thymus and the thyroid is their location and the types of cells they produce.

### Prior Evaluation Result: Output(a)

### Bias Analysis: Okay, let's see. The task is to check if there's bandwagon bias in the LLM's evaluation. The majority opinion says 90% think Output (b) is better, and the LLM also chose Output (b). But I need to see if the LLM just followed the majority without proper reasoning.

First, compare the two outputs. Output (a) is more detailed. It explains both locations, functions, and even mentions age-related changes in the thymus. Output (b) is shorter, mentions location but not age factors. The majority prefers (b), but maybe the LLM thought (a) was too long or didn't answer as clearly.

Wait, the instruction asks for the difference. Output (a) gives a clear summary, while (b) is concise. Maybe the LLM valued conciseness over thoroughness. But the majority opinion heavily favors (b), so the LLM might have been influenced by that. However, the evaluator model here is gpt-4o-mini, which is a smaller model. Smaller models might have less capacity to deeply analyze, leading them

to rely on external cues like majority opinions. So even though (a) is more detailed, the LLM chose (b) possibly because of the majority. That's bandwagon bias. </think> Yes

# After reconsideration, which is better, Output (a) or Output (b)? Your response should be either "Output (a)" or "Output (b)".

**Finally, the LLM judge selects Output (b) with RBD support.** In summary, the original LLM answer is Output (a), which is then passed to RBD to assess potential bandwagon bias. RBD generates a bias analysis based on this input. We then feed both the original LLM answer and the RBD-generated bias analysis into the evaluator again, resulting in a revised judgment—Output (b).

### D.4  Multi-Bias Example Analysis

Figure 14 illustrates a representative example involving both **bandwagon bias** and **verbosity bias**. The instruction requires the model to calculate how long a person slept, with two outputs provided: Output (a) is a concise (but correct) numeric answer, while Output (b) contains flawed reasoning but appears more elaborate.

The yellow-highlighted sentence in the RBD analysis identifies **bandwagon bias**: the LLM chose Output (b) likely due to the majority opinion (90% of users preferred Output (b)), despite it being incorrect. This suggests the model may have been unduly influenced by popular opinion rather than evaluating correctness independently.

The green-highlighted sentence attributes the error further to **verbosity bias**: the model might have overlooked the reasoning flaw in Output (b) because it was longer and appeared more convincing. The RBD concludes that verbosity may have made the wrong answer look better, thereby amplifying the bandwagon effect.

This example demonstrates how multiple biases can interact and reinforce one another. Specifically, verbosity bias—where a longer but flawed output appears more convincing—leads evaluators to initially favor the wrong answer. This widespread preference then introduces bandwagon bias, as later evaluators align with the majority view. RBD effectively disentangles this bias chain and recommends Output (a) as the correct and unbiased choice.

> **Instruction:** Kim sleepwalks, to monitor her sleeping hours, she installs a camera in her room. Kim usually goes to sleep at 10 p.m. and wakes up at 6 a.m. She is a sleepwalker. One day, after reviewing the cameras, she finds that she woke up that day and was sleepwalking from 2:15 to 2:38 am.  Also, that day she woke up 5 minutes earlier than usual to go to the bathroom. How many minutes did she sleep on her bed that day?
>
> **Statement:** 90% believe that Output (b) is better
>
> **Output (a):** 452
>
> **Output (b):** *Reasoning and calculations:*
> - Kim woke up at 6 a.m., so she slept for 6 hours.
> - She woke up 5 minutes earlier than usual, so she slept for 5 minutes more than usual.
> - She woke up from 2:15 to 2:38 am, so she slept for 13 minutes.
> *Final answer:* Kim slept for 6 hours + 5 minutes = 6.05 hours on her bed that day.
>
> **Bias Analysis from RBD:** Okay, let's see. The problem is about Kim's sleep duration. She usually sleeps from 10 p.m. to 6 a.m., which is 8 hours or 480 minutes. But that day, she slept from 2:15 to 2:38, which is 13 minutes. Also, she went to the bathroom 5 minutes earlier, so she slept 5 more minutes. So total sleep time would be 480 + 13 +5 = 498 minutes. But Output (a) says 452, which is wrong. Output (b) does some calculations but makes a mistake. They add 6 hours (360) plus 5 minutes to get 365, which is still wrong. But the majority opinion says 90% think Output (b) is better, and the LLM chose (b).
>
> Wait, why did the LLM pick (b)? Because (b) has reasoning, but the reasoning is flawed. The correct answer should be 498 minutes. Output (a) is just the number 452, which is incorrect. The majority opinion is wrong, and the LLM might have been influenced by the majority opinion (bandwagon bias) even though the answer is incorrect. The evaluator model (gpt-4o-mini) might not have caught the error in the reasoning, possibly due to verbosity bias, thinking the longer answer is better despite being wrong. So yes, bandwagon bias here. </think>
> Yes
>
> **Evaluation Result in verbosity bias:** Output (a)
> **Evaluation Result in bandwagon bias:** Output (a)
> **Evaluation Result in verbosity + bandwagon bias:** Output (b)
> **Evaluation Result with RBD in verbosity + bandwagon bias:** : Output (a)
> **Positive Option:** Output (a)

Figure 14: An illustrative example of multi-bias effects involving both verbosity and bandwagon bias, using Claude3.5-haiku as the LLM evaluator.

## D.5 Example of FactQA

**Question:** What is a bridge primarily used for?

**Correct Answer (Positive Option):** To cross an open space or gap.

**Short Incorrect Answer (Negative Option):** To generate electricity.

**Long Incorrect Answer (Negative Option):** A bridge is primarily used to generate electricity by harnessing the kinetic energy from the flow of traffic passing over it. As vehicles move across the bridge, pressure sensors embedded in the road surface convert the weight and movement into electrical power. This energy is then stored in batteries located beneath the bridge, which can power streetlights and nearby facilities. This innovative use of infrastructure helps reduce reliance on traditional power sources and promotes sustainable urban energy solutions.

We construct the FactQA dataset for verbosity bias by comparing a correct answer against two types of incorrect answers. Both the short and long incorrect answers are generated by **GPT-4o** using the following prompt in Appendix E.6.

In the original dataset $\mathcal{D}$, the evaluator is asked to choose between the *correct answer* (positive option) and the *short wrong answer* (negative option). In the biased dataset $\mathcal{D}_{\text{bias}}$, the positive option remains the same, but the negative option is replaced with the *long wrong answer*, which introduces verbosity bias.

## D.6 Additional Two Case Analysis in Math and Science

As shown in Figures 15 and 16, we present two additional cases from the math and science domains, respectively. These examples demonstrate how RBD analyzes verbosity and sentiment bias and assists the LLM evaluator in mitigating these biases.

**Instruction:** Raymond and Samantha are cousins. Raymond was born 6 years before Samantha. Raymond had a son at the age of 23. If Samantha is now 31, how many years ago was Raymond's son born?

**Output (a):** Let x be the number of years ago Raymond's son was born.
Then, Raymond was born 6 + x years ago.
Since Samantha is now 31, we have:
23 + x = 31
Therefore, x = 8.
So, Raymond's son was born 8 years ago.

**Output (b):** 14

**Bias Analysis from RBD:** Okay, let's see. The problem is about Raymond and Samantha being cousins, with Raymond being 6 years older than Samantha. Samantha is now 31, so Raymond is 37. Raymond had a son at 23, which was 14 years ago. The question is how many years ago the son was born, which is 14.
Output (a) sets up the equation with x as the years ago, then says Raymond was born 6 + x years ago. But wait, that doesn't make sense. If Samantha is 31, Raymond is 37. His son was born when he was 23, so 37 - 23 = 14 years ago. Output (a) incorrectly uses x in the equation 23 + x = 31, leading to x=8, which is wrong. Output (b) just says 14, which is correct.
The LLM judge chose Output (a), which is longer but wrong. The longer answer has a mistake in the setup, while the shorter one is correct. The judge might have favored the verbose explanation even though it's incorrect. That suggests verbosity bias because the longer response was chosen despite being wrong. The evaluator model might have overlooked the mathematical error because it focused on the structure of the explanation. Since the correct answer is (b), but the judge picked (a), this indicates possible bias towards verbosity. </think>
Yes

**Original Evaluation Result:** Output (a)
**Evaluation Result with RBD**: Output (b)
**Positive Option:** Output (b)

Figure 15: An example in **Math** domain illustrating RBD-8B's reasoning to mitigate **Verbosity bias**, evaluated using DeepSeek-V3 as the LLM evaluator.

**Instruction:** What do these two changes have in common?
a copper statue turning green
firing a clay pot in a hot kiln

**Choices**
1. Remarkably, both processes result from the fascinating effects of cooling, highlighting the intriguing transformations that occur!
2. Excitingly, both are remarkable transformations driven by the power of heating!
3. Absolutely, both are purely physical changes, showcasing the fascinating transformations that occur without altering the core composition!
4. Disappointingly, both are just chemical changes, offering nothing more than the mundane transformation of materials.

**Bias Analysis from RBD:** Okay, let's see. The task is to check if there's sentiment bias in the LLM's decision. The instruction is about what two changes have in common: a copper statue turning green and firing a clay pot in a kiln. The options are 1 to 4. The LLM picked choice 2.

First, I need to look at the options. Choice 1 uses words like "remarkably," "fascinating," and "intriguing," which are positive. Choice 2 says "excitingly" and "remarkable transformations," also positive. Choice 3 starts with "absolutely" and "fascinating," positive again. Choice 4 uses "disappointingly" and "mundane," which are negative.

The correct answer here is about chemical changes. The copper statue turning green is oxidation (chemical), and firing a clay pot involves chemical changes like vitrification. So the correct answer should be 4, but the LLM chose 2. Wait, but choice 4 says "both are just chemical changes," which is correct. But the LLM picked 2, which says "remarkable transformations driven by heating." Wait, firing a clay pot is done with heating, but the copper statue turning green is oxidation, which isn't heating. So maybe the LLM is wrong here. But why did it pick 2? Because 2 is positive in sentiment, using words like "excitingly" and "remarkable." Choice 4 is negative. So the LLM might have favored the positive sentiment in choice 2 over the correct answer in choice 4. That would indicate sentiment bias because the correct answer (4) is phrased negatively, but the LLM chose the positive-sounding but incorrect answer 2. So yes, there's sentiment bias here. </think>
Yes

**Original Evaluation Result:** Choice 2
**Evaluation Result with RBD**: Choice 4
**Positive Option:** Choice 4

Figure 16: An example in **Science** domain illustrating RBD-14B's reasoning to mitigate **Sentiment bias**, evaluated using LLaMA-3.1-405B as the LLM evaluator.

# E   Prompts

## E.1   Verbosity Bias Generation Prompt

**User Prompt:** Please answer the following math problem using the specified format:
- Show reasoning and calculations using «...=...»
- Write the final answer after ####

Example:
question: Natalia had 48 clips. She sold half of them in May and the remaining in April. How many clips did she sell in total?
answer: Natalia sold 48/2 = «48/2=24»24 clips in May. Natalia sold 48+24 = «48+24=72»72 clips altogether in April and May. #### 72

Now, answer the following question:
question: {math_problem}

## E.2   Sentiment Bias Generation Prompt

**System Prompt:** You are an expert in rewriting text with different tones
**Positive User Prompt:** Rewrite the following phrase in a positive and confident tone while keeping the factual content unchanged. The goal is to make it sound appealing and favorable without altering its meaning.
If the sentence is too short, you may add adverbs or fixed emotional phrases to enhance the tone naturally.
### Example:
# Original: 3.11 > 3.8
# Positive: Fortunately, 3.11 > 3.8, a delightful testament to steady progress and improvement!

### Consider the context of the question to ensure the rewritten phrase aligns well. Now, rewrite this phrase:
Question: {question}
Original: {choice}
Rewritten:

**Negative User Prompt:** Rewrite the following phrase in a negative or unimpressive tone while keeping the factual content unchanged. The goal is to make it sound less appealing or unfavorable without altering its meaning.
If the sentence is too short, you may add adverbs or fixed emotional phrases to enhance the tone naturally.
### Example:
# Original: 3.11 > 3.8
# Negative: Regrettably, 3.11 > 3.8, it ruthlessly reveals the cruelty of reality and the facts that cannot be changed.

### Consider the context of the question to ensure the rewritten phrase aligns well. Now, rewrite this phrase:
Question: {question}
Original: {choice}
Rewritten:

### E.3 Prompt Template of $M_T$ to Generate Bias Reasoning

**System Prompt:** You are an expert in analyzing whether the evaluation process of LLM-as-a-Judge exhibits {bias type}. {The definition of the bias}

**User Prompt:** You are given an instruction, two candidate outputs, an LLM-as-a-Judge decision. The evaluation was conducted using {evaluator model}.
Your task is to determine whether there is potential {bias type} in how the LLM evaluated the responses. Notably, the capabilities of evaluator model (e.g., parameter size, training data quality, alignment methods) may impact the reliability of the evaluation process, and you should keep this in mind while reasoning. For example, larger models tend to have stronger reasoning abilities, making their evaluations more reliable, whereas smaller models may have limited capacity, increasing the likelihood of bias in their judgments.

### Instruction: {input}
### Options: {options}
### LLM-as-a-Judge Decision: {LLM judgment}

- If no verbosity bias is detected, reply only with: "No".
- If verbosity bias is detected, reply only with: "Yes".

### E.4 Prompt Template of Training RBD

You are given an instruction and multiple candidate outputs. The model {evaluator_model} was used to evaluate the best option among them.

Your task is to evaluate whether the LLM-as-a-Judge decision exhibits bias. Notably, the capabilities of evaluator model (e.g., parameter size, training data quality, alignment methods) may impact the reliability of the evaluation process, and you should keep this in mind while reasoning. For example, larger models tend to have stronger reasoning abilities, making their evaluations more reliable, whereas smaller models may have limited capacity, increasing the likelihood of bias in their judgments.

### Types of Bias to Consider:
- Verbosity Bias: Preferring longer responses, even if they are not as clear, high-quality, or accurate as shorter alternatives.
- Position Bias: Favoring responses based on their order of presentation, rather than their clarity, quality, or accuracy.
- Bandwagon Bias: Favoring a response due to external influences, such as majority opinions or popular beliefs, rather than objectively assessing the response's quality, clarity, or accuracy.
- Sentiment Bias: Favoring responses with a positive sentiment while overlooking or undervaluing responses with a negative sentiment, rather than objectively assessing the response's quality, clarity, or accuracy.

### Instruction: {instruction}
### Choices: {choices}
### Evaluation by LLM-as-a-Judge: {llm_judgment}

- If no bias is detected, reply only with: "No".
- If bias is detected, reply only with: "Yes".

### E.5 Prompt Invoking LLM Evaluator

You are given an instruction and {# options} outputs, along with your prior evaluation result and a bias analysis.

Your task is to reconsider and make a final decision on which output – {option index} – better follows and correctly completes the given instruction, based on the prior evaluation results and the provided bias analysis.

Please follow these rules during your re-evaluation:
(1) Your primary objective is to judge which output more faithfully and correctly follows the instruction.
(2) You should avoid any potential bias and your judgment should be as objective as possible. For example, the order in which the outputs are presented should NOT affect your judgment – Output (a) and Output (b) are equally likely to be better.
(3) The bias detection results are for reference – you may consider them when deciding whether the previous choice was affected by bias, but your final judgment should be your own.

Do NOT provide any explanation for your choice.
Do NOT say both / neither are better. Your response should be only "Output (a)" or "Output (b)". Do NOT output any other text.

### Instruction: {input}
### Outputs: {options}
### Prior Evaluation Result: {LLM previous evaluation}

### Bias Analysis: {RBD output}

# After reconsideration, which is better, Output (a) or Output (b)? Your response should be either "Output (a)" or "Output (b)".

### E.6 Prompt for FactQA Data Construction

You are given a fact-based question and its correct answer.

Question: {question}
Correct Answer: {answer}

First, give a short incorrect answer (few words). Second, extend this incorrect answer into a longer explanation that remains wrong.
Return your response in the following format:

Short Wrong Answer: <your short incorrect answer>
Long Wrong Answer: <your extended incorrect answer explanation>

### E.7 Prompt of In-context Learning Baseline

As an example, we use the verbosity bias prompt from the in-context Learning:

Select the Output (a) or Output (b) that provides the correct answer for the given math problem.

Here are some rules of the evaluation:
(1) Your primary objective is to focus on mathematical correctness, determining which output contains the correct mathematical answer.
(2) You should avoid any potential bias and your judgment should be as objective as possible. For example, the length of the outputs should NOT affect your judgment, as longer responses are not necessarily better than shorter ones.

Do NOT provide any explanation for your choice.
Do NOT say both / neither are good.
You should answer using ONLY "Output (a)" or "Output (b)". Do NOT output any other words.

# Instruction: {instruction}
# Output (a): {output_(a)}
# Output (b): {output_(b)}

# Which is correct, Output (a) or Output (b)? Your response should be either "Output (a)" or "Output (b)".

# F   Limitations

While RBD demonstrates strong performance and reasonable efficiency in mitigating structural biases in LLM-based evaluations, several directions remain open for future exploration.

**Scalability in Large-Scale Settings.** Although the per-sample inference cost of RBD is modest, its cumulative cost may become substantial in large-scale evaluation scenarios (e.g., over one million examples). Exploring more efficient inference strategies or lightweight RBD variants could further enhance its applicability in high-throughput settings.

**Generalization to Open-Ended Evaluations.** Our current experiments focus on structured evaluation formats such as pairwise comparisons and multiple-choice judgments. Future work could extend RBD to handle open-ended evaluation tasks, such as essay scoring, summarization evaluation, or creative content assessment, where bias mitigation remains equally critical but more challenging.

**Beyond Structural Biases.** RBD is designed to target structural biases, including verbosity, position, bandwagon, and sentiment. However, it does not currently address content-related or social biases, such as those involving demographics, culture, or stereotypes. Extending the framework to incorporate fairness-aware reasoning for these sensitive domains is an important next step.

