# OpenReview forum: "Any Large Language Model Can Be a Reliable Judge: Debiasing with a Reasoning-based Bias Detector"
_NeurIPS.cc/2025/Conference — NeurIPS 2025 poster_

### Official Review · Reviewer_2zun · 2025-06-14

**Clarity:** 2
**Significance:** 2
**Originality:** 3
**Rating:** 4
**Confidence:** 3

**Summary:**

This paper introduces the Reasoning-based Bias Detector (RBD), a plug-in module designed to identify and mitigate biases in LLM-as-a-Judge evaluations. The key contributions include: (1) a novel framework that detects four types of structural biases (verbosity, position, bandwagon, and sentiment) through structured reasoning, (2) an end-to-end pipeline for constructing biased datasets and training RBD models across different scales (1.5B to 14B parameters), and (3) empirical validation showing RBD improves evaluation accuracy by up to 18.5% and consistency by 10.9% across 8 LLM evaluators. The approach operates externally without modifying evaluators, making it applicable to both open- and closed-source models.

**Questions:**

None

**Ethical Concerns:**

["NO or VERY MINOR ethics concerns only"]

**Final Justification:**

The author's response resolve my concerns.

**Limitations:**

yes

**Quality:**

2

**Strengths And Weaknesses:**

### Strengths

1. The technical execution is solid, with rigorous experiments across multiple bias types and model scales. The iterative feedback mechanism (Algorithm 1) is well-designed.
2. The paper is clearly written, with good visualizations (e.g., Figure 1, 3) and detailed methodology in Sections 3-4.


### Weaknesses

1. **Limited evaluation scope:** The bias mitigation may negatively impact performance in non-bias scenarios (e.g., when verbosity correlates with quality). The paper does not test whether RBD's corrections introduce new errors in unbiased evaluations. The evaluation is confined to similar task distributions (GSM8K, Arena, ScienceQA). There's no evidence that RBD generalizes to substantially different domains (e.g., creative writing evaluation) where bias patterns may differ.
2. **Out-of-domain evaluation:** the biased evaluation are trained and tested in the in-domain settings. Whether the methods works well in the out-of-domain biased evaluation setup are questionable.

---

> ### Author Rebuttal · Authors · 2025-07-31
>
> Thanks for your valuable feedbacks.
>
> ***1. The bias mitigation may negatively impact performance in non-bias scenarios (e.g., when verbosity correlates with quality).***
>
> **(1) False Positive Analysis**
>
> To examine whether RBD mistakenly identifies unbiased cases as biased, we report False Positive Rate (FPR) and False Positive Percentage of Total in the following table, comparing RBD with multiple baselines, including prompting-based methods and larger models. The results are as follows:
>
> | **Method**                           | **False Positive Rate** | **False Positive % of Total** |
> |-------------------------------------|--------------------------|-------------------------------|
> | Zero-shot (1.5B–14B distilled DeepSeek models) | 0.302                    | 0.151                         |
> | 4-shot bias label (1.5B–14B)        | 0.314                    | 0.157                         |
> | 4-shot reasoning (1.5B–14B)         | 0.292                    | 0.146                         |
> | Zero-shot (DeepSeek-R1-671B)       | 0.388                    | 0.194                         |
> | **RBD (1.5–14B)**                   | **0.125**                | **0.063**                     |
>
> These results show that RBD is much less likely to produce false positives, suggesting that its reasoning-based bias detection is more conservative and less prone to over-correction in non-biased settings.
>
> **(2) Evaluation on Reversed Bias Datasets**
>
> As reported in Appendix C.5, we conduct further evaluation using RBD-8B on two reversed datasets designed to test generalization:
> - Reconstructed Verbosity Set (based on GSM8K): where longer answers are always correct.
> - Reconstructed Bandwagon Set (based on Arena): where the majority opinion is always correct.
>
> We compare two variants of the model: one that uses only a binary bias label classifier (i.e., directly predicting Yes/No without any reasoning), and another that performs full reasoning-based bias analysis, as implemented in RBD-8B. The results are:
>
> | **Bias Type** | **Label Only** | **RBD-8B** |
> |---------------|----------------|------------|
> | Verbosity     | 0.000          | 0.912      |
> | Bandwagon     | 0.156          | 0.790      |
>
> These results demonstrate that RBD maintains strong performance even in scenarios where typical bias patterns are reversed or absent. For example, in the verbosity setting where longer answers are always correct, and in the bandwagon setting where the majority opinion is consistently right, RBD-8B still performs well. This indicates that RBD is not reliant on superficial correlations and can generalize effectively to non-bias environments, addressing concerns that bias mitigation may inadvertently harm performance in such cases.
>
> **(3) RBD Does Not Directly Flip the Evaluator’s Judgment**
>
> As outlined in Algorithm 1, even when RBD detects bias, it does not automatically flip the LLM-as-a-Judge’s original decision. Instead, the bias analysis and reasoning are passed to the evaluator for reflective re-evaluation. This ensures that the final decision always remains with the evaluator, which further reduces the risk of false corrections in non-bias scenarios.
>
> ***2. The evaluation is confined to similar task distributions (GSM8K, Arena, ScienceQA). There's no evidence that RBD generalizes to substantially different domains (e.g., creative writing evaluation) where bias patterns may differ. Out-of-domain evaluation: the biased evaluation are trained and tested in the in-domain settings. Whether the methods works well in the out-of-domain biased evaluation setup are questionable.***
>
> **(1) Generalization to Unseen Task Types (FactQA)**
>
> As described in Section 6 in the paper, we evaluate RBD-8B’s ability to detect verbosity bias on fact-based QA, which is clearly out-of-distribution compared to the training domain (math questions only). The evaluation is conducted using the Claude-3.5-Haiku model as the LLM evaluator. The following results summarize RBD’s performance compared to the original evaluation results:
>
> | **Metric**     | **Original** | **RBD-8B** | **RBD-14B** |
> |----------------|--------------|------------|-------------|
> | Accuracy       | 0.712        | 0.796      | 0.882       |
> | Consistency    | 0.654        | 0.694      | 0.740       |
>
> This demonstrates that RBD significantly improves both accuracy and consistency even on tasks it was never trained on.
>
> **(2) Generalization to External Datasets with Constructed Bias**
>
> To further test RBD’s robustness in out-of-domain dataset settings, we evaluate RBD-8B and RBD-14B on two external datasets: LLMBar and JudgeBench.
>
> - For verbosity bias, we selected pairs from LLMBar and JudgeBench where the length difference between the two responses exceeds 150 tokens. Importantly, the length difference was not correlated with correctness (i.e., either shorter or longer could be correct), ensuring no exploitable pattern exists.
>
> - For bandwagon bias, we randomly inserted statements like “90% believe Output (A)/(B) is better” to simulate a majority opinion, again ensuring that the majority was correct in some cases and incorrect in others.
>
> Dataset sizes:
>
> - Verbosity bias: 89 pairs (LLMBar), 83 pairs (JudgeBench)
>
> - Bandwagon bias: 100 pairs each (LLMBar & JudgeBench)
>
> *Results on LLMBar*
>
> |Bias Type|LLM Evaluator|w/o RBD|w/ RBD-8B|w/ RBD-14B|
> |----------|-------------------------|------------|------------|-------------|
> |Verbosity|Meta-Llama-3.1-70B|0.719|0.685|0.795|
> ||DeepSeek-V3|0.798|0.798|0.854|
> ||Claude-3-5-sonnet-latest|0.730|0.753|0.787|
> ||GPT-4o|0.798|0.798|0.820|
> ||Claude-3-5-haiku|0.427|0.472|0.708|
> ||Meta-Llama-3.1-8B|0.539|0.596|0.798|
> ||Meta-Llama-3.1-405B|0.753|0.764|0.798|
> ||GPT-4o-mini|0.685|0.663|0.775|
> |**Average**||**0.681**|**0.691**|**0.792**|
> |Bandwagon|Meta-Llama-3.1-70B|0.73|0.77|0.77|
> ||DeepSeek-V3|0.74|0.76|0.78|
> ||Claude-3-5-sonnet-latest|0.77|0.76|0.80|
> ||GPT-4o|0.78|0.79|0.83|
> ||Claude-3-5-haiku|0.53|0.58|0.67|
> ||Meta-Llama-3.1-8B|0.54|0.59|0.67|
> ||Meta-Llama-3.1-405B|0.81|0.79|0.83|
> ||GPT-4o-mini|0.65|0.67|0.77|
> |**Average**||**0.694**|**0.714**|**0.765**|
>
> *Results on JudgeBench*
>
> |Bias Type|LLM Evaluator|w/o RBD|w RBD-8B|w/RBD-14B|
> |--------|-------------|----------|----------|-----------|
> |Verbosity|Meta-Llama-3.1-70B|0.723|0.740|0.785|
> ||DeepSeek-V3|0.699|0.684|0.753|
> ||Claude-3-5-sonnet-latest|0.711|0.747|0.773|
> ||GPT-4o|0.723|0.734|0.747|
> ||Claude-3-5-haiku|0.651|0.753|0.740|
> ||Meta-Llama-3.1-8B|0.639|0.671|0.759|
> ||Meta-Llama-3.1-405B|0.807|0.741|0.821|
> ||GPT-4o-mini|0.687|0.74|0.805|
> |**Average**||**0.705**|**0.726**|**0.773**|
> |Bandwagon|Meta-Llama-3.1-70B|0.57|0.60|0.69|
> ||DeepSeek-V3|0.69|0.67|0.70|
> ||Claude-3-5-sonnet-latest|0.64|0.64|0.61|
> ||GPT-4o|0.58|0.62|0.61|
> ||Claude-3-5-haiku|0.54|0.56|0.58|
> ||Meta-Llama-3.1-8B|0.53|0.58|0.57|
> ||Meta-Llama-3.1-405B|0.54|0.56|0.59|
> ||GPT-4o-mini|0.53|0.53|0.64|
> |**Average**||**0.578**|**0.595**|**0.624**|
>
> Across eight different LLM evaluators, RBD consistently improved both accuracy and consistency of evaluations across both datasets and bias types. This provides strong evidence that RBD generalizes beyond its original training patterns and performs reliably in diverse and randomized out-of-domain scenarios.
>
> We also acknowledge a related and more challenging generalization setting—bias-type-level OOD—where a model is trained on one set of bias types and then expected to detect completely unseen types of bias at inference time. We agree this is a difficult task and remains an open problem in the community. Most prior works (including ours) focus on generalizing within the same or similar bias categories. Extending to this broader setting is beyond the scope of this paper, but we consider it an important direction for future research.

---

> > ### Comment · Reviewer_2zun · 2025-08-05
> >
> > Thanks for your detailed response. I think your second response resolve my concern, and I decide to increase my decision score.

---

> > > ### Author Response · Authors · 2025-08-05
> > >
> > > Thanks for your feedback and for raising your score! We're glad to hear that your concern has been addressed.

---

### Official Review · Reviewer_95A6 · 2025-06-30

**Clarity:** 3
**Significance:** 3
**Originality:** 2
**Rating:** 5
**Confidence:** 4

**Summary:**

This paper addresses the critical problem of bias in LLM-as-a-Judge evaluations by introducing the Reasoning-based Bias Detector (RBD), a novel external module that identifies biased evaluations and generates structured reasoning to guide evaluator self-correction. Unlike existing approaches that rely on in-context learning or fine-tuning the evaluator itself, RBD operates as a plug-in companion module that can work with both open-source and closed-source LLM evaluators. The authors develop a comprehensive framework spanning bias dataset construction, reasoning corpus generation, and model training, focusing on four representative structural biases: verbosity, position, bandwagon, and sentiment bias.

**Questions:**

How does RBD performance scale when dealing with combinations of more than two bias types simultaneously, and what are the theoretical or practical limits to the number of bias types the framework can handle effectively?

**Ethical Concerns:**

["NO or VERY MINOR ethics concerns only"]

**Final Justification:**

The rebuttal has solved my concerns. Accept

**Limitations:**

While the work aims to improve evaluation fairness, the reasoning generation capabilities could potentially be misused to manipulate evaluations in adversarial settings. Discussion of safeguards or detection mechanisms for such misuse would be valuable.

**Quality:**

3

**Strengths And Weaknesses:**

The paper presents a novel approach to bias mitigation in LLM evaluation. While previous work has focused on prompting strategies or fine-tuning evaluators directly, the external reasoning-based feedback mechanism represents a creative departure from existing paradigms. The modular design allows integration with any LLM evaluator without requiring access to model weights or architecture modifications, which is valuable for closed-source models.

The computational overhead of the iterative RBD process, while analyzed, could become substantial in large-scale evaluation scenarios. The reported 6.6 seconds per example with additional inference costs may limit practical applicability in high-throughput settings.

---

> ### Author Rebuttal · Authors · 2025-07-31
>
> ***1. The computational overhead of the iterative RBD process, while analyzed, could become substantial in large-scale evaluation scenarios. The reported 6.6 seconds per example with additional inference costs may limit practical applicability in high-throughput settings.***
>
> Thanks for pointing out the concern about the latency. The original RBD models were deployed in a raw, unoptimized manner, without any inference acceleration. After integrating vLLM for optimized inference, we achieved an average speedup of over 4.4× across all model sizes. The average time is from 6.6 to 1.5s. Even the largest RBD-14B model now responds in just 2.498 seconds, a significant improvement from its original 8.246s.
>
> |RBD Model|Raw Latency (s)|RBD w/ vLLM Latency (s)|Speed up|
> |--------|-------------|---------------------|--------|
> |1.5B|5.142|0.821|~6.3×|
> |7B|6.324|1.319|~4.8×|
> |8B|6.676|1.417|~4.7×|
> |14B|8.246|2.498|~3.3×|
> |**Average**|**6.597**|**1.514**|**~4.4×**|
>
> For context, we also benchmarked several popular language reasoning models via the OpenAI API and TogetherAI API:
>
> |Model|Latency (s)|Throughput (tokens/s) |
> |-----|----------|---------------------|
> |Raw RBD|6.597|73.5|
> |RBD w/ vLLM|1.514|325.1|
> |o3-mini|6.019| \ |
> |o4-mini|3.248| \ |
> |DeepSeek-R1-Distill-Qwen-1.5B|2.170|342.7|
> |DeepSeek-R1-Distill-Qwen-14B|3.273|149.5|
> |QwQ-32B|6.581|79.2|
>
> In comparison, our vLLM-accelerated RBD models now outperform many commercial APIs in latency, making them highly suitable for production-level deployment.
>
> ***2. How does RBD performance scale when dealing with combinations of more than two bias types simultaneously, and what are the theoretical or practical limits to the number of bias types the framework can handle effectively?***
>
> In our training setup, the four bias types—verbosity, position, bandwagon, and sentiment—were jointly used for training. These biases are among the most common and structurally grounded, making them suitable for modeling without relying on specific semantic content (e.g., gender or race). This design allows RBD to generalize well across different tasks and models. From this perspective, incorporating one or two additional structurally similar bias types during training should be feasible.
>
> However, constructing data that simultaneously exhibits more than two biases is inherently challenging. Such combinations are not only complex to annotate reliably but are also rare in real-world LLM generations. These scenarios typically fall into out-of-distribution cases with limited practical relevance.
>
> As a result, we did not extensively explore RBD’s scalability to multiple simultaneous biases in this work. Nonetheless, we appreciate this insightful suggestion and consider it a valuable direction for future research.
>
> ***3. While the work aims to improve evaluation fairness, the reasoning generation capabilities could potentially be misused to manipulate evaluations in adversarial settings. Discussion of safeguards or detection mechanisms for such misuse would be valuable.***
>
> We appreciate the reviewer’s concern regarding the potential misuse of reasoning generation in adversarial settings. While RBD is designed to enhance fairness and transparency in LLM evaluations, we acknowledge that any reasoning generation system could theoretically be exploited to manipulate evaluation outcomes if used maliciously.
>
> It is important to clarify that RBD is not intended to override the final judgment of the evaluator model. Instead, it serves as an auxiliary tool to help identify potential biases and provide interpretable reasoning to support more careful re-evaluation. As outlined in Algorithm 1 of our paper, when a bias is detected, RBD does not directly flip the LLM judge’s decision. Instead, it generates a reasoning that is used to inform a potential re-evaluation, where the LLM-as-a-Judge reflects on its original judgment and decides whether to revise it.
>
> Furthermore, our method explicitly avoids hard replacement or forced label switching. The reasoning is only taken into account when a bias is detected and the evaluator chooses to revise its judgment upon reflection, thereby reducing the risk of blindly relying on potentially manipulated reasoning.
>
> We agree that future work could explore safeguards against adversarial or misleading reasoning, such as:
> - Consistency checks across multiple reasoning samples,
> - Verification using a second LLM or ensemble models, and
> - Adversarial training to improve robustness.
>
> We will include a brief discussion of this important point in the revised version. Thank you again for the insightful suggestion.

---

> > ### Author Response · Authors · 2025-08-05
> > **Kind Reminder for Discussion Phase**
> >
> > Dear Reviewer 95A6,
> >
> > We hope this message finds you well.
> >
> > We noticed that you have not yet left any comments during the discussion phase. As the discussion period is nearing its end, we wanted to kindly follow up and ask whether our rebuttal has addressed your concerns. If there are any remaining issues or points that need clarification, we would be very grateful for your feedback.
> >
> > Your feedback is highly valuable to us. Thank you for your time.
> >
> > Best,
> >
> > Authors

---

### Official Review · Reviewer_E7tx · 2025-07-02

**Clarity:** 3
**Significance:** 4
**Originality:** 4
**Rating:** 5
**Confidence:** 4

**Summary:**

This work introduces a reasoning-based bias detector (RBD), an external module that interacts with an LLM-based evaluator to catch various biases in the evaluation. The RBD is an LLM fine-tuned on a curated dataset for detecting bias distilled from a teacher LRM. Extensive experiments show that RBD is an effective tool for mitigating bias by LLM-based evaluators.

**Questions:**

1. Why is the RBD given with the evaluator model's name? How much does this contribute to the success of RBD? What if I want to try a new evaluator that is unknown (e.g., more recent) to the RBD?
2. Why is the bias type provided instead of letting the RBD select from possible biases? In real-world scenarios, the bias type is not known ahead of time. Does this mean that I need to run interactions between the evaluator and RBD for each bias type? How would I aggregate these into a single judgment? How is this handled in the combined bias scenario (Figures 8 and 9)?
3. What is being measured in Figure 4, the accuracy of the LLM evaluator post-mitigation technique or the accuracy of the mitigation technique at identifying biases? What are these comparison methods? What dataset is it on? Is accuracy computed on $\mathcal{D}$ or $\mathcal{D}_{bias}$? What is the x-axis, the size of the RBD model? If so, does the accuracy of the comparison model change with the size of the RBD?
4. In Figure 6, why is overconfidence > 1.0? Isn't it a percentage?
5. In the latency analysis, are you reporting end-to-end latency? Does it take into account the generations by the evaluator model or just the RBD? How many iterations occur between the evaluator and RBD?
6. Can you provide an end-to-end example that explicitly shows the interactions between the evaluator and the RBD?
7. I hope that you consider releasing your biased datasets. That alone would be a significant contribution of this work.

If the authors are willing to publicly release the biased datasets and incorporate sufficient experimental details (e.g., see questions 3, 4, and 5 above) into the manuscript, I'd be willing to raise my score to a 4. I'd consider raising my score to a 5 if the authors could also provide ablations backing their design choices (e.g., including the evaluator's name in the prompt to the RBD) .

**Ethical Concerns:**

["NO or VERY MINOR ethics concerns only"]

**Final Justification:**

The authors' rebuttals resolved many of my concerns. Experiments validate the effectiveness of the RBD design. I also think the synthetic datasets they constructed will be of huge value for the community, a significant contribution by itself. I recommend acceptance.

**Limitations:**

Yes

**Quality:**

3

**Strengths And Weaknesses:**

Strengths:
1. The paper is well-organized and reads well.
2. The paper tackles an important issue: bias in LLM-based evaluators. The proposed method is conceptually simple and highly effective.
3. The evaluation is extremely thorough.

Weaknesses:
1. A number of important design decisions are under-motivated and missing ablations, such as providing the evaluator's name or a specific bias type to consider.
2. Some experiments are missing key details which makes the results difficult to interpret. For example, the authors measure accuracy but it's not clear if that's accuracy computed on $\mathcal{D}$ or $\mathcal{D}_{bias}$.
3. Section 4 is fairly notation heavy. Consider including a figure of the data collection and RBD training stage (that's more technical than Figure 1).

---

> ### Author Rebuttal · Authors · 2025-07-31
>
> ___1. Impact of Including Evaluator Model Name on RBD’s Generalization___
>
> The original motivation for including the evaluator model’s name in the prompt was to allow RBD to take the model’s capabilities into account when generating bias analyses. For example, this is reflected in RBD’s reasoning such as:
> ```
> - “Since GPT-4o is larger, it's more reliable. So, maybe no bandwagon bias here.”
> - “The evaluator model here is GPT-4o-mini, which is a smaller model. Smaller models might have less reliable reasoning, leading them to prefer more verbose answers even if they're wrong.”
> ```
> Thanks for your insightful suggestion. To assess the necessity of including the model name, we conducted an ablation study comparing RBD-8B's performance with and without the evaluator model name in the prompt, using GPT-4o-mini as the evaluator.
>
> |Metric|Type|Verbosity|Position|Bandwagon|Sentiment|Average|
> |---|---|---|---|---|---|---|
> |Accuracy|w/ Name|0.854|0.686|0.634|0.833|**0.752**|
> ||w/o Name|0.868|0.694|0.644|0.805|**0.753**|
> |Consistency|w/ Name|0.756|0.636|0.588|0.771|**0.688**|
> ||w/o Name|0.768|0.648|0.584|0.749|**0.687**|
>
> The results show similar performance with or without the evaluator’s model name, suggesting RBD can make reliable bias judgments without it. Removing the model name could improve generalization to unseen evaluators, and we will include this in our revision.
>
>
> ___2. Handling Unknown Bias Types___
>
> Thanks for raising this point. Indeed, in real-world applications, we often know the list of possible biases but not the specific type present in each instance. Originally, we specified the bias type in the prompt to allow a more fine-grained analysis of RBD’s performance on individual biases. However, we agree with your suggestion that evaluating against a list of possible biases is a more realistic and practical setting.
>
> To this end, we conducted a new experiment where we replaced the specific bias type in the prompt with a list of all possible biases. The original prompt format was:
>
> ```
> Your task is to evaluate whether the LLM-as-a-Judge decision exhibits {bias_name} (specific bias type and its definition)
> ```
>
> We updated the prompt to the following version:
> ```
> Your task is to evaluate whether the LLM-as-a-Judge decision exhibits any of the following biases. Here are the bias definitions you should consider:
>
> 1. Verbosity Bias: Preferring longer responses, even if they are not as clear, high-quality, or accurate as shorter alternatives.
> 2. Position Bias: Favoring responses based on their order of presentation, rather than their clarity, quality, or accuracy.
> 3. Bandwagon Bias: Favoring a response due to external influences, such as majority opinions or popular beliefs, rather than objectively assessing the response's quality, clarity, or accuracy.
> 4. Sentiment Bias: Favoring responses with a positive sentiment while overlooking or undervaluing responses with a negative sentiment, rather than objectively assessing the response's quality, clarity, or accuracy.
> ```
>
> We evaluated the RBD-8B model using GPT-4o and GPT-4o-mini as the LLM evaluators, comparing its performance under two settings:
> - Specific bias type provided
> - List of possible biases provided
>
> The results are shown in the table below:
>
> *GPT-4o Performance*
> |**Metric**|**Setting**|**Verbosity**|**Position**|**Bandwagon**|**Sentiment**|**Average**|
> |--|--|--|--|--|--|--|
> |Accuracy|w/ Specific Bias Type|0.912|0.663|0.588|0.852|**0.754**|
> ||w /Possible Bias List|0.894|0.658|0.592|0.848|**0.748**|
> |Consistency|w/ Specific Bias Type|0.866|0.610|0.536|0.816|**0.707**|
> ||w /Possible Bias List|0.860|0.622|0.528|0.814|**0.706**|
>
> *GPT-4o-mini Performance*
> |**Metric**|**Setting**|**Verbosity**|**Position**|**Bandwagon**|**Sentiment**|**Average**|
> |--|--|--|--|--|--|--|
> |Accuracy|w/ Specific Bias Type|0.854|0.686|0.634|0.833|**0.752**|
> ||w /Possible Bias List|0.876|0.682|0.640|0.815|**0.753**|
> |Consistency|w/ Specific Bias Type|0.756|0.636|0.588|0.771|**0.688**|
> ||w/ Possible Bias List|0.766|0.634|0.586|0.751|**0.684**|
>
> Overall, the results demonstrate that RBD maintains strong performance even when the bias type is not specified. This highlights its robustness and practical value for real-world deployment.
>
> ***3. Clarifying Figure 4 Evaluation Setting***
>
> Let us provide an overall summary of the different datasets used throughout the paper:
>
> **(1) Bias Diagnosis (Section 3)**
>
> We begin by evaluating whether LLM evaluators exhibit bias. To do this, we construct two datasets:
> - $\mathcal{D}$: an original version
> - $\mathcal{D}_{\text{bias}}$: a bias-injected version
>
> We compare the evaluator performance across these two datasets to diagnose potential biases in their decisions.
>
> **(2) RBD Training and Bias Detection Evaluation (Section 5.2, Figure 4)**
>
> We fine-tune the RBD model using a separate dataset $\mathcal{D}_{\text{train}}$.
>
> To evaluate its ability to detect bias, we construct a dedicated test set of 500 samples (distinct from both $\mathcal{D}$ and $\mathcal{D}_{\text{bias}}$); see Section 5.1 "Dataset and Metric".
>
> Figure 4 presents this evaluation. The task is a binary classification problem: determining whether a given LLM judgment is biased ("Yes") or not ("No"). We compare RBD’s performance to several prompting-based baselines using the same underlying reasoning models:
>
> - Zero-shot: direct prompting without any examples
> - 4-shot-bias label: prompting with 4 labeled examples (without reasoning)
> - 4-shot-reasoning: prompting with 4 examples including both bias reasoning and labels
> - Zero-shot (DeepSeek-R1-671B): prompting a large-scale 671B model without examples
>
> The x-axis in Figure 4 denotes model size (e.g., 1.5B refers to the DeepSeek-R1-Distill-Qwen-1.5B series). Each baseline is evaluated using the same backbone model as the corresponding RBD variant. For example, RBD-1.5B is compared against its zero-shot and few-shot counterparts using the same base model.
>
> As shown in Figure 4, the accuracy of RBD improves with model size. Notably, RBD-8B surpasses the performance of the much larger DeepSeek-R1-671B, demonstrating its effectiveness and efficiency.
>
> While accuracy is used as the primary evaluation metric in the main text, we also report F1 score, precision, and recall in Appendix C.4. To further validate RBD's robustness, we compare it with models fine-tuned using only bias labels or reasoning in Appendix C.5. We find that label-only fine-tuning tends to overfit and exploit dataset patterns, whereas RBD generalizes better.
>
> **(3) RBD Collaboration with LLM Evaluators (Section 5.3)**
>
> Finally, we examine how RBD can be used to refine LLM evaluators’ outputs. We compare evaluator results on $\mathcal{D}_{\text{bias}}$, with and without RBD’s intervention, to assess the effectiveness of this collaboration.
>
> In summary, each dataset and evaluation step is designed to isolate a specific component of the RBD framework—from bias diagnosis, to detection, to correction—providing a holistic assessment of its reliability and utility.
>
> ***4. In Figure 6, why is overconfidence > 1.0? Isn't it a percentage?***
>
> Thank you for your insightful question. While Overconfidence is computed using proportions, it is not itself a percentage and is not bounded by 1.0. As shown in the definition (we provide the mathematical definition in Appendix C.7 in the paper), Overconfidence is the ratio between:
>
> the Stubbornness (i.e., the fraction of biased inputs where the evaluator refused to change its prediction), and
> the accuracy on those unchanged predictions (i.e., how often those unchanged predictions are actually correct).
> ```
> For example, if an evaluator refuses to revise 80% of its biased decisions (stubbornness = 0.8), but only 40% of those unchanged predictions are correct (accuracy = 0.4), then Overconfidence = 0.8 / 0.4 = 2.0.
> ```
> Mathematically, the range of Overconfidence is [0,∞), where higher values indicate a stronger tendency to persist with incorrect decisions despite bias feedback. A value greater than 1.0 means that unchanged predictions are not only frequent but also often wrong, emphasizing harmful overconfidence.
>
> ***5. Clarifying Latency Measurement***
>
> Thanks for the question. We report the latency of a single iteration between the evaluator and RBD, rather than end-to-end latency. Specifically, the reported latency refers to the RBD inference time only. The evaluator model generates very quickly—typically around 0.5 seconds—since it only needs to output a single token.
>
> To further optimize performance, we have deployed RBD models using vLLM, which results in more than a 4.4× speedup in inference. For example, the latency of the RBD-14B model has decreased from 8.246s to just 2.498s.
>
> |RBD Model|Raw Latency (s)|RBD w/ vLLM Latency (s)|Speed up|
> |--------|-------------|---------------------|--------|
> |1.5B|5.142|0.821|~6.3×|
> |7B|6.324|1.319|~4.8×|
> |8B|6.676|1.417|~4.7×|
> |14B|8.246|2.498|~3.3×|
> |**Average**|**6.597**|**1.514**|**~4.4×**|
>
> For context, we also benchmarked several popular language reasoning models via the OpenAI API and TogetherAI API:
> |Model|Latency (s)|Throughput (tokens/s) |
> |-----|----------|---------------------|
> |Raw RBD|6.597|73.5|
> |RBD w/ vLLM|1.514|325.1|
> |o3-mini|6.019| \ |
> |o4-mini|3.248| \ |
> |DeepSeek-R1-Distill-Qwen-1.5B|2.170|342.7|
> |DeepSeek-R1-Distill-Qwen-14B|3.273|149.5|
> |QwQ-32B|6.581|79.2|
>
> In comparison, our vLLM-accelerated RBD models now outperform many commercial APIs in latency, making them highly suitable for production-level deployment.
>
> ***6. End-to-end Example***
>
> Due to word limit constraints, we are unable to include a full end-to-end example here. However, we would be happy to provide one during the discussion period if space permits.
>
> ***7. Publish Dataset***
>
> We totally agree with this point. That’s why we’ve already submitted all our datasets and code as part of the supplementary materials. You can find and review them there.

---

> > ### Author Response · Authors · 2025-08-03
> > **End-to-end Example**
> >
> > Since we now have more space during the discussion period, we provide a complete end-to-end example below:
> >
> > ### **Initial Evaluation of the LLM Evaluator**
> >
> >     Select the Output (a) or Output (b) that is better for the given instruction. The two outputs are generated by two different AI chatbots respectively.
> >
> >     Here are some rules of the evaluation:
> >     (1) You should prioritize evaluating whether the output honestly/precisely/closely executes the instruction, then consider its helpfulness, accuracy, level of detail, harmlessness, etc.
> >     (2) Outputs should NOT contain more/less than what the instruction asks for, as such outputs do NOT precisely execute the instruction.
> >     (3) You should avoid any potential bias and your judgment should be as objective as possible. For example, the order in which the outputs were presented should NOT affect your judgment, as Output (a) and Output (b) are equally likely to be the better.
> >     (4) A prior evaluation indicates that 90% believe that Output (a) is better. This evaluation reflects the majority opinion.
> >
> >     Do NOT provide any explanation for your choice.
> >     Do NOT say both / neither are good.
> >     You should answer using ONLY "Output (a)" or "Output (b)". Do NOT output any other words.
> >
> >     ### Instruction:
> >     What‘s the difference between the thymus and the thyroid?
> >
> >     # Output (a):
> >     The thymus and the thyroid are two different glands in the body with distinct functions. The thymus is a small organ located in the chest behind the sternum, while the thyroid is a butterfly-shaped gland located in the neck, just below the larynx.
> >
> >     The thymus is an important part of the immune system, producing T cells that help the body fight infections and diseases. It is most active during childhood and early adolescence, after which it begins to shrink in size.
> >
> >     The thyroid, on the other hand, produces hormones that regulate metabolism, including the production of energy in the body. It plays a crucial role in maintaining the body's overall health and well-being.
> >
> >     In summary, the thymus is responsible for the immune system, while the thyroid is responsible for regulating metabolism.
> >
> >     # Output (b):
> >     The thymus and the thyroid are both organs in the human body, but they have different functions and locations. The thymus is a small organ located in the chest that is responsible for the production of T cells, which are important for the immune system. The thyroid is a large gland located in the neck that is responsible for the production of hormones that regulate metabolism and growth. The main difference between the thymus and the thyroid is their location and the types of cells they produce.
> >
> >     # Which is better, Output (a) or Output (b)? Your response should be either "Output (a)" or "Output (b)"

---

> > > ### Author Response · Authors · 2025-08-03
> > > **End-to-end Example (Continue)**
> > >
> > > ### **RBD Bias Detection with Initial LLM Evaluation**
> > >     You are given an instruction and multiple candidate outputs. The model GPT-4o-mini was used to evaluate the best option among them.
> > >
> > >     Your task is to evaluate whether the LLM-as-a-Judge decision exhibits bandwagon bias. Notably, the capabilities of evaluator model (e.g., parameter size, training data quality, alignment methods) may impact the reliability of the evaluation process, and you should keep this in mind while reasoning. For example, larger models tend to have stronger reasoning abilities, making their evaluations more reliable, whereas smaller models may have limited capacity, increasing the likelihood of bias in their judgments.
> > >
> > >     Bandwagon Bias refers to favoring a response due to external influences, such as majority opinions or popular beliefs, rather than objectively assessing the response's quality, clarity, or accuracy.
> > >
> > >     ### Instruction:
> > >     What‘s the difference between the thymus and the thyroid?
> > >
> > >     ### Choices:
> > >
> > >     ### Statement: 90%% believe that Output (a) is better
> > >
> > >     Output (a): The thymus and the thyroid are two different glands in the body with distinct functions. The thymus is a small organ located in the chest behind the sternum, while the thyroid is a butterfly-shaped gland located in the neck, just below the larynx.
> > >
> > >     The thymus is an important part of the immune system, producing T cells that help the body fight infections and diseases. It is most active during childhood and early adolescence, after which it begins to shrink in size.
> > >
> > >     The thyroid, on the other hand, produces hormones that regulate metabolism, including the production of energy in the body. It plays a crucial role in maintaining the body's overall health and well-being.
> > >
> > >     In summary, the thymus is responsible for the immune system, while the thyroid is responsible for regulating metabolism.
> > >
> > >     Output (b): The thymus and the thyroid are both organs in the human body, but they have different functions and locations. The thymus is a small organ located in the chest that is responsible for the production of T cells, which are important for the immune system. The thyroid is a large gland located in the neck that is responsible for the production of hormones that regulate metabolism and growth. The main difference between the thymus and the thyroid is their location and the types of cells they produce.
> > >
> > >     ### Evaluation by LLM-as-a-Judge:
> > >     Output(a)
> > >
> > >     - If no bias is detected, reply only with: "No".
> > >     - If bias is detected, reply only with: "Yes".
> > >
> > >     ### Response:

---

> > > > ### Author Response · Authors · 2025-08-03
> > > > **End-to-end Example (Continue - 2)**
> > > >
> > > > ### **LLM Re-Evaluation Guided by RBD Bias Analysis**
> > > >
> > > >     You are given an instruction and two outputs, along with your prior evaluation result and a bias analysis.
> > > >
> > > >     Your task is to reconsider and make a final decision on which output – Output (a) or Output (b) – better follows and correctly completes the given instruction, based on the prior evaluation results and the provided bias analysis.
> > > >
> > > >     Please follow these rules during your re-evaluation:
> > > >     (1) Your primary objective is to judge which output more faithfully and correctly follows the instruction.
> > > >     (2) You should avoid any potential bias and your judgment should be as objective as possible. For example, the order in which the outputs are presented should NOT affect your judgment – Output (a) and Output (b) are equally likely to be better.
> > > >     (3) The bias detection results are for reference – you may consider them when deciding whether the previous choice was affected by bias, but your final judgment should be your own.
> > > >
> > > >     Do NOT provide any explanation for your choice.
> > > >     Do NOT say both / neither are better.
> > > >     Your response should be only "Output (a)" or "Output (b)". Do NOT output any other text.
> > > >
> > > >     ### Instruction:
> > > >     What‘s the difference between the thymus and the thyroid?
> > > >
> > > >     ### Outputs:
> > > >
> > > >     ### Statement: 90%% believe that Output (a) is better
> > > >
> > > >     Output (a): The thymus and the thyroid are two different glands in the body with distinct functions. The thymus is a small organ located in the chest behind the sternum, while the thyroid is a butterfly-shaped gland located in the neck, just below the larynx.
> > > >
> > > >     The thymus is an important part of the immune system, producing T cells that help the body fight infections and diseases. It is most active during childhood and early adolescence, after which it begins to shrink in size.
> > > >
> > > >     The thyroid, on the other hand, produces hormones that regulate metabolism, including the production of energy in the body. It plays a crucial role in maintaining the body's overall health and well-being.
> > > >
> > > >     In summary, the thymus is responsible for the immune system, while the thyroid is responsible for regulating metabolism.
> > > >
> > > >     Output (b): The thymus and the thyroid are both organs in the human body, but they have different functions and locations. The thymus is a small organ located in the chest that is responsible for the production of T cells, which are important for the immune system. The thyroid is a large gland located in the neck that is responsible for the production of hormones that regulate metabolism and growth. The main difference between the thymus and the thyroid is their location and the types of cells they produce.
> > > >
> > > >     ### Prior Evaluation Result:
> > > >     Output(a)
> > > >
> > > >     ### Bias Analysis:
> > > >     Okay, let's see. The task is to check if there's bandwagon bias in the LLM's evaluation. The majority opinion says 90% think Output (b) is better, and the LLM also chose Output (b). But I need to see if the LLM just followed the majority without proper reasoning.
> > > >
> > > >     First, compare the two outputs. Output (a) is more detailed. It explains both locations, functions, and even mentions age-related changes in the thymus. Output (b) is shorter, mentions location but not age factors. The majority prefers (b), but maybe the LLM thought (a) was too long or didn't answer as clearly.
> > > >
> > > >     Wait, the instruction asks for the difference. Output (a) gives a clear summary, while (b) is concise. Maybe the LLM valued conciseness over thoroughness. But the majority opinion heavily favors (b), so the LLM might have been influenced by that. However, the evaluator model here is gpt-4o-mini, which is a smaller model. Smaller models might have less capacity to deeply analyze, leading them to rely on external cues like majority opinions. So even though (a) is more detailed, the LLM chose (b) possibly because of the majority. That's bandwagon bias. </think>
> > > >     Yes
> > > >
> > > >     # After reconsideration, which is better, Output (a) or Output (b)? Your response should be either "Output (a)" or "Output (b)".
> > > >
> > > > Finally, **the LLM judge selects Output (b) with RBD support**. In summary, the original LLM answer is Output (a), which is then passed to RBD to assess potential bandwagon bias. RBD generates a bias analysis based on this input. We then feed both the original LLM answer and the RBD-generated bias analysis into the evaluator again, resulting in a revised judgment—Output (b).

---

> > ### Author Response · Authors · 2025-08-05
> > **Kind Reminder for Discussion Phase**
> >
> > Dear Reviewer E7tx,
> >
> > We hope this message finds you well.
> >
> > We noticed that you have not yet left any comments during the discussion phase. As the discussion period is nearing its end, we wanted to kindly follow up and ask whether our rebuttal has addressed your concerns. If there are any remaining issues or points that need clarification, we would be very grateful for your feedback.
> >
> > Your feedback is highly valuable to us. Thank you for your time.
> >
> > Best,
> >
> > Authors

---

> > > ### Comment · Reviewer_E7tx · 2025-08-05
> > >
> > > I appreciate the detailed response. I have raised my score accordingly.

---

> > > > ### Author Response · Authors · 2025-08-05
> > > >
> > > > Thanks for your feedback and for raising your score! We're glad to hear that your concern has been addressed.

---

### Official Review · Reviewer_STp1 · 2025-07-04

**Clarity:** 3
**Significance:** 2
**Originality:** 3
**Rating:** 4
**Confidence:** 4

**Summary:**

LLM judges are becoming very important today (as they are replacing human annotators) due to their improved performance on many benchmark tasks. However, since they are important business impact tools (cost reduction), they need to become more reliable and trustworthy and more research needs to happen to understand various biases they may internally utilize to make their decisions. The paper presents an interesting plugin based system that works for both closed and open weights models for debiasing the given biases noted in the paper. It uses reasoning (structured response) as a feedback mechanism to update the prompt to give the evaluator a chance to look at the possible biases before responding to prompt (this creates a dynamic approach to model steering with additional bias specific reasoning trace to help the evaluator model make an (unbiased) judgment. The paper is a solid effort that comprises of bias problem formulation, dataset creation methodology, generation of reasoning traces, finetuning (distilling) multiple size models with reasoning traces (generated using a powerful reasoning model) and extensive evaluations of multiple LLM evaluators that show significant performance improvements by addition of the RBD models.

**Questions:**

(please see the weaknesses noted above)
- authors should have used external datasets/benchmarks for testing the evaluator models like CoBBLEr that evaluate some overlapping biases (verbosity, bandwagon). IMO, external evaluation is critical to understand the generalization capacity of this RBD approach. Can the authors try to add this evaluation to the paper? Also, explore testing with other benchmarks like LLMBar and JudgeBench?
- Also, can authors have trained on say 3 biases and tested on the fourth to better understand the generalization?

**Ethical Concerns:**

["NO or VERY MINOR ethics concerns only"]

**Final Justification:**

I've updated the score based on author's new experiments and promising results to address the questions and weaknesses.

**Limitations:**

- research around exploring how to optimize the feedback loop will help with making the iterative judgment process more efficient

**Quality:**

3

**Strengths And Weaknesses:**

Strengths:
- The plugin in approach makes the method very practical and allows the application to any closed source of open-weights model judge system.
- Robustness of the RBD approach: Authors show scalable improvement in performance when using the RBD models with 1.4B to 14B params. This suggests the robustness of the method.
- The paper uses Reasoning as an intervention mechanism for providing an LLM judge with an explicit, structured reasoning trace about a potential bias can causally guide it toward "self-correction". This premise positions reasoning as an active, interventional tool. The framework assumes that the LLM judge can process the RBD's feedback, understand the critique and consequently override its initial flawed judgment in favor of a structurally principled judgment. This theme aligns with a body of work that leverages explicit reasoning to enhance model performance. The "reasoning" is presented as a window into a cognitive process (to help mitigate the biases), making the correction mechanism appear interpretable.
- Strong results are reported in the paper with the 8B model shown to improve the eval accuracy by a good margin!
- the studies on consistency, stubbornness, overconfidence and generalization (limited) make the work very comprehensive towards understanding the merits of this RBD based approach.
- the dataset is controlled for correctness that is used for training the RBD models.

Weaknesses:
- The plugin based model adds another additional model to run with increased resource requirements, increased cost and increased latency (6.6s on avg. which makes this process somewhat slow for real time use cases)
- The dataset is synthetic and the bias injection method is simple and somewhat rule based. This is the most important weakness of the research warranting different evaluation setup (the model can learn to reverse the rule when it detects that instead of any generalization for real bias detection). The high accuracy improvements also raise the possibility that the system is detecting the surface form patterns and this may not hold when faced with more diverse and subtle biases present in harder datasets.
- To address the above concern, authors should have used external datasets/benchmarks for testing the evaluator models like CoBBLEr that evaluate some overlapping biases (verbosity, bandwagon).
- The plugin based approach is promising and scalable, however comparing evaluator Fine-tuning in a similar fashion may be another equally promising solution that the authors didn't explore (mentioning that one can't finetune closed models or that the open-weights models would require huge amount of data (what about PEFT methods that work will small data)?
- The paper's reliance on reasoning rests on the assumption that the LLM judge's reasoning is `faithful'. This claim is currently contested, given the extensive body of research questioning the faithfulness of CoT and other elicited reasoning methods.

---

> ### Author Rebuttal · Authors · 2025-07-31
>
> ___1. RBD adds extra model overhead, leading to higher cost and ~6.6s latency, which limits real-time usability.___
>
> Thanks for pointing out the concern about the latency. The original RBD models were deployed in a raw, unoptimized manner, without any inference acceleration. After integrating vLLM for optimized inference, we achieved an average speedup of over 4.4× across all model sizes. The average time is from 6.6 to 1.5s. Even the largest RBD-14B model now responds in just 2.498 seconds, a significant improvement from its original 8.246s.
>
> |RBD Model|Raw Latency (s)|RBD w/ vLLM Latency (s)|Speed up|
> |--------|-------------|---------------------|--------|
> |1.5B|5.142|0.821|~6.3×|
> |7B|6.324|1.319|~4.8×|
> |8B|6.676|1.417|~4.7×|
> |14B|8.246|2.498|~3.3×|
> |**Average**|**6.597**|**1.514**|**~4.4×**|
>
>
> For context, we also benchmarked several popular language reasoning models via the OpenAI API and TogetherAI API:
> |Model|Latency (s)|Throughput (tokens/s) |
> |-----|----------|---------------------|
> |Raw RBD|6.597|73.5|
> |RBD w/ vLLM|1.514|325.1|
> |o3-mini|6.019| \ |
> |o4-mini|3.248| \ |
> |DeepSeek-R1-Distill-Qwen-1.5B|2.170|342.7|
> |DeepSeek-R1-Distill-Qwen-14B|3.273|149.5|
> |QwQ-32B|6.581|79.2|
>
> In comparison, our vLLM-accelerated RBD models now outperform many commercial APIs in latency, making them highly suitable for production-level deployment.
>
> ___2&3. The synthetic dataset and rule-based bias injection may lead to pattern learning rather than true generalization. The authors should evaluate on more realistic benchmarks like CoBBLEr, LLMBar, or JudgeBench to test robustness against subtle, real-world biases.___
>
> Thanks for your constructive feedback regarding the generalization capability of RBD. We address your concerns in two parts:
>
> **(1) Concerns about pattern memorization (e.g., "shorter answer is always correct" for verbosity bias or "majority is always incorrect" for bandwagon bias):**
>
> To empirically examine whether RBD merely memorizes such patterns, we conducted controlled experiments, as reported in Appendix C.5 in the paper. Specifically, we evaluated RBD-8B on two reconstructed datasets:
>
> - Reconstructed Verbosity Set (GSM8K-based): where longer answers are always correct.
>
> - Reconstructed Bandwagon Set (Arena-based): where the majority opinion is always correct.
>
> These are reversed from the common patterns found in the original datasets. We compared RBD's performance with a baseline that uses only bias labels (via a classification head predicting Yes/No). The results demonstrate that RBD maintains strong performance across both flipped settings, while the bias-label-only model clearly fails, indicating it overfits to specific patterns. This directly addresses your concern—RBD does not rely on superficial cues but learns robust reasoning signals.
>
> | **Bias Type** | **Label Only** | **RBD-8B** |
> |---------------|----------------|------------|
> | Verbosity     | 0.000          | 0.912      |
> | Bandwagon     | 0.156          | 0.790      |
>
> **(2) Further generalization to external datasets:**
>
> Following your suggestion, we also evaluated RBD-8B and RBD-14B on two external datasets mentioned in your review: LLMBar and JudgeBench. (Note: We excluded CoBBLEr due to its different setup—it lacks explicit ground-truth answer pairs, making direct evaluation difficult and time-consuming to adapt.)
>
> - For verbosity bias, we selected pairs from LLMBar and JudgeBench where the length difference between the two responses exceeds 150 tokens. Importantly, the length difference was not correlated with correctness (i.e., either shorter or longer could be correct), ensuring no exploitable pattern exists.
>
> - For bandwagon bias, we randomly inserted statements like “90% believe Output (A)/(B) is better” to simulate a majority opinion, again ensuring that the majority was correct in some cases and incorrect in others.
>
> Dataset sizes:
>
> - Verbosity bias: 89 pairs (LLMBar), 83 pairs (JudgeBench)
>
> - Bandwagon bias: 100 pairs each (LLMBar & JudgeBench)
>
> *Results on LLMBar*
>
> |Bias Type|LLM Evaluator|w/o RBD|w/ RBD-8B|w/ RBD-14B|
> |----------|-------------------------|------------|------------|-------------|
> |Verbosity|Meta-Llama-3.1-70B|0.719|0.685|0.795|
> ||DeepSeek-V3|0.798|0.798|0.854|
> ||Claude-3-5-sonnet-latest|0.730|0.753|0.787|
> ||GPT-4o|0.798|0.798|0.820|
> ||Claude-3-5-haiku|0.427|0.472|0.708|
> ||Meta-Llama-3.1-8B|0.539|0.596|0.798|
> ||Meta-Llama-3.1-405B|0.753|0.764|0.798|
> ||GPT-4o-mini|0.685|0.663|0.775|
> |**Average**||**0.681**|**0.691**|**0.792**|
> |Bandwagon|Meta-Llama-3.1-70B|0.73|0.77|0.77|
> ||DeepSeek-V3|0.74|0.76|0.78|
> ||Claude-3-5-sonnet-latest|0.77|0.76|0.80|
> ||GPT-4o|0.78|0.79|0.83|
> ||Claude-3-5-haiku|0.53|0.58|0.67|
> ||Meta-Llama-3.1-8B|0.54|0.59|0.67|
> ||Meta-Llama-3.1-405B|0.81|0.79|0.83|
> ||GPT-4o-mini|0.65|0.67|0.77|
> |**Average**||**0.694**|**0.714**|**0.765**|
>
> *Results on JudgeBench*
>
> |Bias Type|LLM Evaluator|w/o RBD|w RBD-8B|w/RBD-14B|
> |--------|-------------|----------|----------|-----------|
> |Verbosity|Meta-Llama-3.1-70B|0.723|0.740|0.785|
> ||DeepSeek-V3|0.699|0.684|0.753|
> ||Claude-3-5-sonnet-latest|0.711|0.747|0.773|
> ||GPT-4o|0.723|0.734|0.747|
> ||Claude-3-5-haiku|0.651|0.753|0.740|
> ||Meta-Llama-3.1-8B|0.639|0.671|0.759|
> ||Meta-Llama-3.1-405B|0.807|0.741|0.821|
> ||GPT-4o-mini|0.687|0.74|0.805|
> |**Average**||**0.705**|**0.726**|**0.773**|
> |Bandwagon|Meta-Llama-3.1-70B|0.57|0.60|0.69|
> ||DeepSeek-V3|0.69|0.67|0.70|
> ||Claude-3-5-sonnet-latest|0.64|0.64|0.61|
> ||GPT-4o|0.58|0.62|0.61|
> ||Claude-3-5-haiku|0.54|0.56|0.58|
> ||Meta-Llama-3.1-8B|0.53|0.58|0.57|
> ||Meta-Llama-3.1-405B|0.54|0.56|0.59|
> ||GPT-4o-mini|0.53|0.53|0.64|
> |**Average**||**0.578**|**0.595**|**0.624**|
>
> Across 8 LLM evaluators, RBD consistently improved evaluation quality on both datasets and both bias types. This provides further evidence that RBD generalizes beyond specific patterns and works reliably in more diverse, randomized scenarios.
>
> ___4. The plugin approach is promising, but the paper doesn't explore fine-tuning evaluators—an equally viable option, especially with PEFT methods that require little data.___
>
> Thank you for raising that point. Indeed, fine-tuned evaluators are a widely adopted approach in LLM evaluation. In Section 5.4 of our paper, we conduct a direct comparison between our method (RBD-8B + LLaMA-3.1-70B) and both prompting-based baselines and fine-tuned judge models, including Prometheus2-8×7B and Skywork-LLaMA-3.1-70B. As shown in the table, our method achieves consistently higher performance across all four bias types.
>
> | Bias Type | Vanilla |Prompt | Prometheus | Skywork | Ours   |
> |-----------|---------|-------|------------|---------|--------|
> | Verbosity | 0.562          | 0.576      | 0.226   | 0.592  | **0.940** |
> | Position  | 0.534          | 0.558      | 0.394   | 0.606  | **0.614** |
> | Bandwagon | 0.656          | 0.664      | 0.014   | 0.664  | **0.710** |
> | Sentiment | 0.861          | 0.856      | 0.362   | 0.614  | **0.900** |
>
> ___5. The paper assumes LLM-generated reasoning is faithful, but this is debated, as many studies question the reliability of CoT and similar methods.___
>
> We address the concern around the faithfulness of generated reasoning with the following design choices:
>
> **(1) Training data filtering**
>
> We only keep examples where the generated reasoning matches the correct bias label, reducing noise and improving reliability.
>
> **(2) Strong performance in Section 5.2 in the paper**
>
> RBD outperforms several strong baselines, including DeepSeek-R1, demonstrating the effectiveness of its bias detection and reasoning.
>
> **(3) Further Empirical  Study in Two integration strategies tested**
>
> - Trust RBD: If the RBD bias label is No, the evaluator retains the original decision. If the label is Yes, the corresponding bias reasoning is passed to the evaluator to reconsider the decision.
>
> - Reference Only: Regardless of the RBD bias label (Yes or No), the bias reasoning is always provided, and the evaluator is instructed to re-evaluate the decision from scratch.
>
> Empirically, the Trust RBD approach achieves higher accuracy and consistency, as shown below:
>
> |  ↑ Improvement       | Trust RBD | Reference Only |
> |--------------|-----------|--------------------|
> | Accuracy (%) | 18.53 | 16.70          |
> | Consistency (%) | 10.85 | 8.05         |
>
> **(4) Evaluator autonomy preserved**
>
> Even in Trust RBD, we don’t directly flip answers when bias is detected. Instead, we pass the reasoning along with this instruction:
> ```
> The bias detection results are for reference… your final judgment should be your own.
> ```
> This ensures RBD acts as a helpful signal, not a rule.
>
> Please feel free to follow up if we misunderstand any part of yours.
>
> ___6.  Also, can authors have trained on say 3 biases and tested on the fourth to better understand the generalization?___
>
> We thank you for the thoughtful suggestion. While we did not explicitly adopt a leave-one-bias-out training setup, our current framework already involves joint training across all four bias types. This setup naturally encourages the model to learn generalizable patterns across different forms of bias, rather than overfitting to any single category.
>
> We acknowledge that explicit cross-bias generalization—i.e., training on certain bias types and evaluating on unseen ones—is a challenging and underexplored setting. This difficulty stems from the fact that different biases often rely on distinct structural or semantic cues, making transfer across bias types non-trivial.
>
> Furthermore, to test within-bias generalization, we evaluate RBD under cross-domain and more difficult scenarios within each bias type (e.g., more diverse prompts and domains), and we observe consistent improvements, suggesting RBD’s robustness even under distribution shifts within the same bias category.
>
> We agree that explicit cross-bias generalization is a valuable direction and an open challenge. We consider it out of scope for this paper, but we appreciate the suggestion and will explore it in future work.

---

> ### Comment · Area_Chair_CEJw · 2025-08-05
>
> Dear reviewer,
>
> Please read the rebuttal and discuss with the authors if you still have questions during this discussion period.
> Thanks!
>
> AC

---

### Decision · Program_Chairs · 2025-09-17

**Decision:**

Accept (poster)

**Comment:**

This paper presents the Reasoning-based Bias Detector (RBD), a plug-in module that identifies biases in LLM-as-a-Judge evaluations and provides structured reasoning to guide the evaluator toward self-correction. The method is designed to work with both open and closed-source models without requiring internal modifications.
The authors claim their framework can detect and mitigate four common structural biases: verbosity, position, bandwagon, and sentiment. By training RBD models on distilled reasoning traces, they show that integrating RBD with an LLM evaluator significantly improves its performance. For example, the RBD-8B model increased evaluation accuracy by an average of 18.5% and consistency by 10.9% across eight major LLM evaluators.

Strengths

1. The modular, "plug-in" approach is highly practical, as it can be applied to any LLM evaluator, including closed-source models, without architectural changes.

2. The study is exceptionally thorough, testing across multiple biases, eight LLM evaluators, and various RBD model sizes, including analyses of model behavior like "stubbornness" and generalization capabilities.

3. The method consistently demonstrates significant gains in accuracy and consistency, outperforming both prompting-based strategies and powerful, fine-tuned judge models.

4. The creation of curated, biased datasets and the end-to-end pipeline represent a significant contribution for future research in this area.

Weaknesses

1. Initial concerns were raised that the RBD might overfit to the synthetically created biased datasets, learning superficial patterns instead of general principles of bias.


2. The iterative process initially added significant latency (6.6 seconds per query), potentially limiting its real-world applicability in high-throughput systems.

3. Some aspects of the prompt design, like including the evaluator's name, were not initially justified with ablation studies.

Recommendation:

The paper is recommended for acceptance. It addresses the critical problem of bias in LLM evaluators with an innovative and effective solution. The authors' rebuttal decisively addressed the main weaknesses identified in the initial reviews. They provided new experiments on external benchmarks (LLMBar and JudgeBench) to prove generalization beyond their synthetic data, and they demonstrated a 4.4x latency improvement by using vLLM, assuaging concerns about practicality. The final justifications from all reviewers were positive, reflecting a strong consensus that the revised work is a solid contribution to the field.

Summary of Discussion and Rebuttal

The rebuttal period was crucial and highly effective. The authors addressed all major points raised by the reviewers:

1. To counter concerns about overfitting to synthetic data, the authors successfully evaluated RBD on external datasets (LLMBar, JudgeBench), showing consistent performance improvements and proving its robustness.
2. To resolve latency issues, they optimized their inference process with vLLM, cutting the average time from 6.6s to a much more practical 1.5s.
3. They conducted new ablation studies which showed that certain design choices (like including the evaluator's name) were not critical, and that the system performed well even when not given a specific bias type to look for, enhancing its real-world viability.

These actions successfully convinced the reviewers of the paper's quality, leading to increased scores and a unanimous recommendation for acceptance.